# Skd3 (human ClpB) is a potent mitochondrial protein disaggregase that is inactivated by 3-methylglutaconic aciduria-linked mutations

Ryan R Cupo[1,2], James Shorter[1,2]*

[1]Department of Biochemistry and Biophysics, Perelman School of Medicine at the University of Pennsylvania, Philadelphia, United States; [2]Pharmacology Graduate Group, Perelman School of Medicine at the University of Pennsylvania, Philadelphia, United States

**Abstract** Cells have evolved specialized protein disaggregases to reverse toxic protein aggregation and restore protein functionality. In nonmetazoan eukaryotes, the AAA+ disaggregase Hsp78 resolubilizes and reactivates proteins in mitochondria. Curiously, metazoa lack Hsp78. Hence, whether metazoan mitochondria reactivate aggregated proteins is unknown. Here, we establish that a mitochondrial AAA+ protein, Skd3 (human ClpB), couples ATP hydrolysis to protein disaggregation and reactivation. The Skd3 ankyrin-repeat domain combines with conserved AAA+ elements to enable stand-alone disaggregase activity. A mitochondrial inner-membrane protease, PARL, removes an autoinhibitory peptide from Skd3 to greatly enhance disaggregase activity. Indeed, PARL-activated Skd3 solubilizes α-synuclein fibrils connected to Parkinson's disease. Human cells lacking Skd3 exhibit reduced solubility of various mitochondrial proteins, including anti-apoptotic Hax1. Importantly, Skd3 variants linked to 3-methylglutaconic aciduria, a severe mitochondrial disorder, display diminished disaggregase activity (but not always reduced ATPase activity), which predicts disease severity. Thus, Skd3 is a potent protein disaggregase critical for human health.

***For correspondence:**
jshorter@pennmedicine.upenn.edu

**Competing interests:** The authors declare that no competing interests exist.

## Introduction

Protein aggregation and aberrant phase transitions are elicited by a variety of cellular stressors and can be highly toxic (*Chuang et al., 2018*; *Eisele et al., 2015*; *Guo et al., 2019*). To counter this challenge, cells have evolved specialized protein disaggregases to reverse aggregation and restore resolubilized proteins to native structure and function (*Shorter, 2017*; *Shorter and Southworth, 2019*). Indeed, protein disaggregases are conserved across all domains of life, with orthologues of Hsp104, a ring-shaped hexameric AAA+ protein, powering protein disaggregation and reactivation (as opposed to degradation) in eubacteria and nonmetazoan eukaryotes (*Glover and Lindquist, 1998*; *Goloubinoff et al., 1999*; *Queitsch et al., 2000*; *Shorter, 2008*). In nonmetazoan eukaryotes, Hsp104 functions in the cytoplasm and nucleus (*Parsell et al., 1994*; *Tkach and Glover, 2008*; *Wallace et al., 2015*), whereas the closely-related AAA+ disaggregase, Hsp78, resolubilizes and reactivates proteins in mitochondria (*Krzewska et al., 2001*; *Schmitt et al., 1996*). Curiously, at the evolutionary transition from protozoa to metazoa both Hsp104 and Hsp78 are lost and are subsequently absent from all animal species (*Erives and Fassler, 2015*). This loss of Hsp104 and Hsp78 is perplexing given that toxic protein aggregation remains a major challenge in metazoa (*Eisele et al., 2015*). Indeed, it is even more baffling since ectopic expression of Hsp104 is well tolerated by animal cells and can be neuroprotective in animal models of neurodegenerative disease (*Cushman-

*Nick et al., 2013*; *Dandoy-Dron et al., 2006*; *Jackrel et al., 2014*; *Lo Bianco et al., 2008*; *Perrin et al., 2007*; *Satyal et al., 2000*; *Vacher et al., 2005*; *Yasuda et al., 2017*).

Metazoa may partially compensate for the absence of Hsp104 activity in the cytoplasm and nucleus with alternative general protein-disaggregase systems, such as Hsp110, Hsp70, Hsp40, and small heat-shock proteins (*Duennwald et al., 2012*; *Mattoo et al., 2013*; *Nillegoda et al., 2015*; *Scior et al., 2018*; *Shorter, 2011*; *Shorter, 2017*; *Torrente and Shorter, 2013*) as well as client-specific disaggregases in the cytoplasm such as nuclear-import receptors (*Guo et al., 2019*; *Guo et al., 2018*; *Niaki et al., 2020*; *Yoshizawa et al., 2018*). However, Hsp110 is not found in mitochondria (*Voos and Röttgers, 2002*). Thus, it continues to remain uncertain whether, in the absence of Hsp78, metazoan mitochondria harbor a disaggregase that solubilizes and reactivates aggregated proteins.

Here, we investigate whether Skd3 (encoded by human *CLPB*) might act as a mitochondrial protein disaggregase in metazoa (*Figure 1a*). Skd3 is a ubiquitously expressed, mitochondrial AAA+ protein of poorly-defined function, which is related to Hsp104 and Hsp78 via its HCLR clade AAA+ domain (*Figure 1a* and *Figure 1—figure supplement 1*; *Erzberger and Berger, 2006*; *Périer et al., 1995*; *Seraphim and Houry, 2020*). Skd3 appears to play an important role in maintaining mitochondrial structure and function (*Chen et al., 2019*). Curiously, Skd3 first appears in evolution alongside Hsp104 and Hsp78 in choanoflagellates, a group of free-living unicellular and colonial flagellate eukaryotes that are the closest extant protozoan relatives of animals (*Figure 1b* and *Supplementary file 1*; *Brunet and King, 2017*; *Erives and Fassler, 2015*). During the complex evolutionary transition from protozoa to metazoa, Skd3 is retained, whereas Hsp104 and Hsp78 are lost (*Erives and Fassler, 2015*). Indeed, Skd3 is conserved in many metazoan lineages (*Figure 1a,b*, *Figure 1—figure supplement 1*, and *Supplementary file 1*; *Erives and Fassler, 2015*).

Skd3 is comprised of a mitochondrial-targeting signal (MTS), followed by a short hydrophobic stretch, an ankyrin-repeat domain (ANK), an AAA+ nucleotide-binding domain (NBD), and a small C-terminal domain (CTD) (*Figure 1a*). The Skd3 NBD closely resembles NBD2 of Hsp104 and Hsp78 (*Figure 1a* and *Figure 1—figure supplement 1*). Aside from this similarity, Skd3 is highly divergent from Hsp104 and Hsp78 (*Figure 1a* and *Figure 1—figure supplement 1*; *Torrente and Shorter, 2013*). For example, Skd3, Hsp104, and Hsp78 all have short CTDs, but these are divergent with the Skd3 CTD being basic compared to the more acidic Hsp104 and Hsp78 CTDs (*Figure 1—figure supplement 1*). Moreover, the other domains in Hsp104 (N-terminal domain [NTD], NBD1, and middle domain [MD]) and Hsp78 (NBD1 and MD) are not found in Skd3 (*Figure 1a* and *Figure 1—figure supplement 1*). In their place, is an ankyrin-repeat domain (*Figure 1a*), which interestingly is an important substrate-binding domain of another protein disaggregase, chloroplast signal recognition particle 43 (cpSRP43) (*Jaru-Ampornpan et al., 2013*; *Jaru-Ampornpan et al., 2010*; *McAvoy et al., 2018*; *Nguyen et al., 2013*).

Importantly, mutations in the Skd3 ankyrin-repeat domain and NBD are linked to the rare, but severe mitochondrial disorder, 3-methylglutaconic aciduria, type VII (MGCA7) (*Capo-Chichi et al., 2015*; *Kanabus et al., 2015*; *Kiykim et al., 2016*; *Pronicka et al., 2017*; *Saunders et al., 2015*; *Wortmann et al., 2016*; *Wortmann et al., 2015*). MGCA7 is an autosomal recessive metabolic disorder that presents with increased levels of 3-methylglutaconic acid (3-MGA), neurologic deterioration, and neutropenia (*Wortmann et al., 2016*). Typically, patients present with infantile onset of a progressive encephalopathy with movement abnormalities and delayed psychomotor development (*Wortmann et al., 2016*). Other common, but variable, phenotypes include cataracts, seizures, and recurrent infections (*Wortmann et al., 2016*). These issues can be severe with afflicted infants typically only living for a few weeks or months (*Wortmann et al., 2016*). Patients may also present with more moderate phenotypes, including neutropenia, hypotonia, spasticity, movement abnormalities, epilepsy, and intellectual disability (*Wortmann et al., 2016*). Mildly affected individuals have no neurological problems, normal life expectancy, but present with neutropenia (*Wortmann et al., 2016*). There is no cure and no effective therapeutics for severe or moderate forms of MGCA7. Moreover, little is known about how Skd3 mutations might cause disease.

Collectively, these various observations concerning Skd3 led us to hypothesize that Skd3 is a metazoan mitochondrial protein disaggregase of key importance for mitochondrial proteostasis. We further hypothesized that MGCA7-associated Skd3 mutations would disrupt disaggregase activity. Our investigation of these hypotheses is detailed below. Briefly, we find that Skd3 is an ATP-

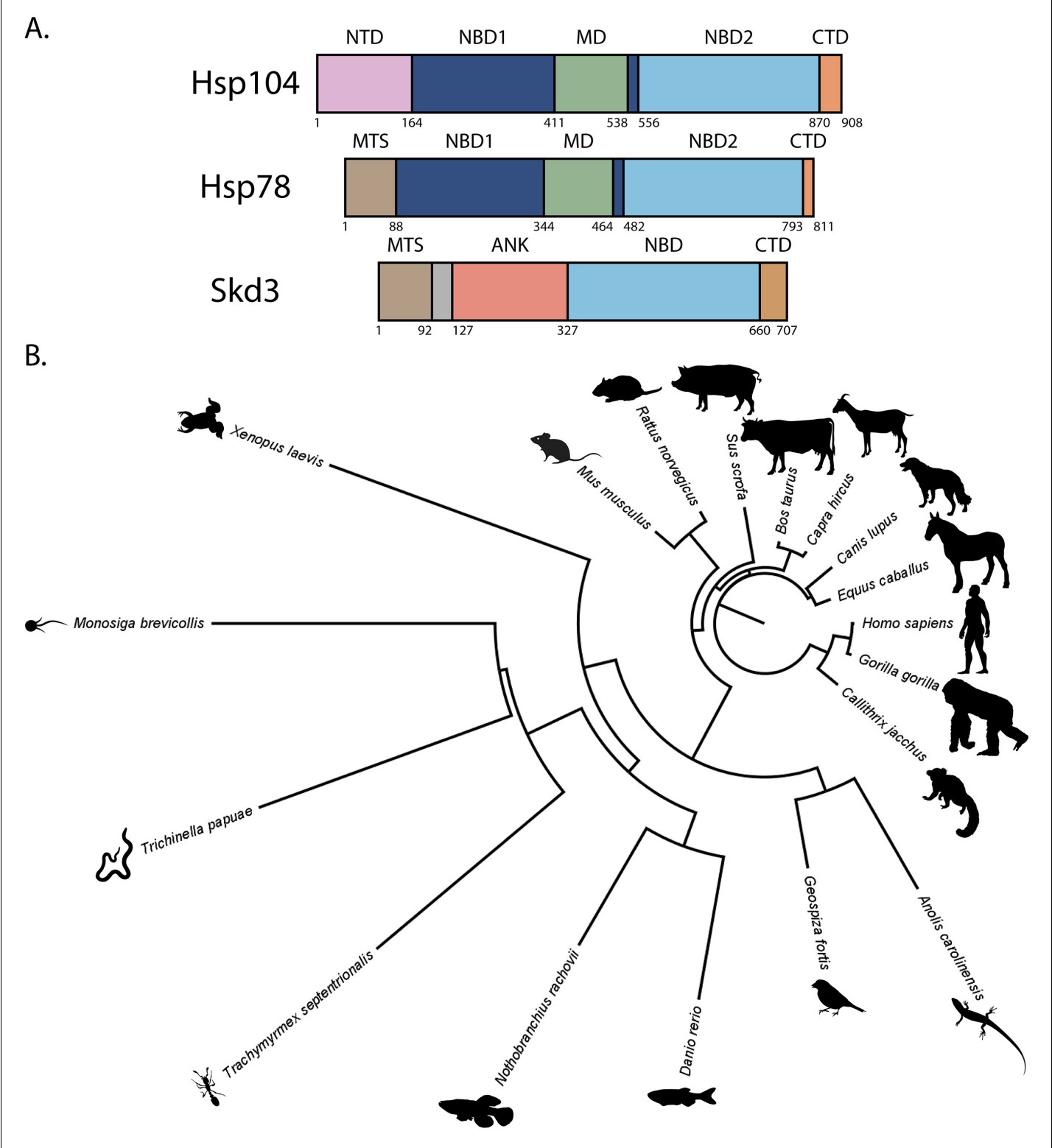

**Figure 1.** Skd3 is homologous to Hsp104 and Hsp78 and is conserved across diverse metazoan lineages. (**A**) Domain map depicting *S. cerevisiae* Hsp104, *S. cerevisiae* Hsp78, and *H. sapiens* Skd3. Hsp104 is composed of a N-terminal domain (NTD), nucleotide-binding domain 1 (NBD1), middle domain (MD), nucleotide-binding domain 2 (NBD2), and C-terminal domain (CTD). Hsp78 is composed of a mitochondrial-targeting signal (MTS), NBD1, MD, NBD2, and CTD. Skd3 is composed of a MTS, a short hydrophobic stretch of unknown function, an ankyrin-repeat domain (ANK) containing four ankyrin repeats, an NBD that is homologous to Hsp104 and Hsp78 NBD2, and a CTD. (**B**) Phylogenetic tree depicting a Clustal Omega alignment

*Figure 1 continued on next page*

*Figure 1 continued*

of Skd3 sequences from divergent metazoan lineages and the protozoan *Monosiga brevicollis*. The alignment shows conservation of Skd3 across diverse species and shows high similarity between mammalian Skd3 proteins.

The online version of this article includes the following figure supplement(s) for figure 1:

**Figure supplement 1.** Skd3 NBD alignment to other AAA+ proteins reveals high similarity to Hsp104 and Hsp78.

dependent mitochondrial protein disaggregase that is activated by the rhomboid protease, PARL, and inactivated by MGCA7-linked mutations.

## Results

### Skd3 couples ATP hydrolysis to protein disaggregation and reactivation

To biochemically dissect the activity of Skd3, we purified full-length Skd3 (see Materials and methods), lacking the mitochondrial targeting signal, which is cleaved by the mitochondrial-processing peptidase (MPP) upon import into mitochondria (*Figure 2a*, *Figure 2—figure supplement 1a–d*; *Claros and Vincens, 1996*; *Wortmann et al., 2015*). We term this form of Skd3, $_{MPP}$Skd3. We first assessed that ATPase activity of $_{MPP}$Skd3 and found that it is active (*Figure 2b* and *Figure 2—figure supplement 2a*). Indeed, $_{MPP}$Skd3 displayed robust ATPase activity that was comparable to Hsp104 and over two-fold higher than previously reported values for $_{MPP}$Skd3 activity (*Figure 2b* and *Figure 2—figure supplement 2a*; *Mróz et al., 2020*; *Wortmann et al., 2015*).

To determine if $_{MPP}$Skd3 is a disaggregase we used a classic aggregated substrate, urea-denatured firefly luciferase aggregates, which form aggregated structures of ~500–2000 kDa and greater in size that are devoid of luciferase activity (*DeSantis et al., 2012*; *Glover and Lindquist, 1998*). Indeed, very few luciferase species smaller than ~400 kDa can be detected (*Glover and Lindquist, 1998*). Importantly, these samples are devoid of misfolded, monomeric luciferase (*Glover and Lindquist, 1998*). Reactivation of luciferase in this assay is thus primarily achieved by the extraction and subsequent refolding of monomeric luciferase ($M_w$ ~61 kDa) from large aggregated structures (~500–2,000 kDa or larger) (*Glover and Lindquist, 1998*). Hence, in this assay, luciferase reactivation is an accurate proxy for luciferase disaggregation. Importantly, Hsp70 and Hsp40 are unable to disaggregate and reactivate luciferase found in these large aggregated structures (~500–2,000 kDa or larger) (*Glover and Lindquist, 1998*). Indeed, Hsp70 and Hsp40 were unable to recover any active luciferase (~0.05 ± 0.00% [mean ± SEM] native luciferase activity) and were similar to the buffer control (~0.04 ± 0.00% native luciferase activity) (*Figure 2c* and *Figure 2—figure supplement 2b,d*). As expected, Hsp104 alone was also unable to disaggregate and reactivate luciferase (~0.08 ± 0.01% native luciferase activity) (*Figure 2c* and *Figure 2—figure supplement 2b,d*; *DeSantis et al., 2012*; *Glover and Lindquist, 1998*). By contrast, the combination of Hsp104, Hsp70, and Hsp40 recovered ~7.68 ± 0.44% native luciferase activity (*Figure 2c* and *Figure 2—figure supplement 2b, d*; *Glover and Lindquist, 1998*). Remarkably, $_{MPP}$Skd3 displayed robust disaggregase activity in the presence of ATP as it was able to recover ~2.87 ± 0.17% native luciferase activity (*Figure 2c* and *Figure 2—figure supplement 2b,c*). Indeed, $_{MPP}$Skd3 displayed ~40% of the disaggregase activity of Hsp104 plus Hsp70 and Hsp40 under these conditions (*Figure 2c*). While Hsp104 required the presence of Hsp70 and Hsp40 to disaggregate luciferase (*Figure 2c* and *Figure 2—figure supplement 2b,c*; *DeSantis et al., 2012*; *Glover and Lindquist, 1998*), $_{MPP}$Skd3 had no requirement for Hsp70 and Hsp40 (*Figure 2c* and *Figure 2—figure supplement 2b,c*). This finding indicates that $_{MPP}$Skd3 is a 'stand-alone' disaggregase.

Next, we assessed the nucleotide requirements for $_{MPP}$Skd3 disaggregase activity. $_{MPP}$Skd3 disaggregase activity was supported by ATP but not by the absence of nucleotide or the presence of ADP (*Figure 2c* and *Figure 2—figure supplement 2b,d*). Likewise, neither the non-hydrolyzable ATP analogue, AMP-PNP, nor the slowly hydrolyzable ATP analogue, ATPγS, could support $_{MPP}$Skd3 disaggregase activity (*Figure 2—figure supplement 2b,d*). Collectively, these data suggest that $_{MPP}$Skd3 disaggregase activity requires multiple rounds of rapid ATP binding and hydrolysis, which is similar mechanistically to Hsp104 (*DeSantis et al., 2012*; *Shorter and Lindquist, 2004*; *Shorter and Lindquist, 2006*).

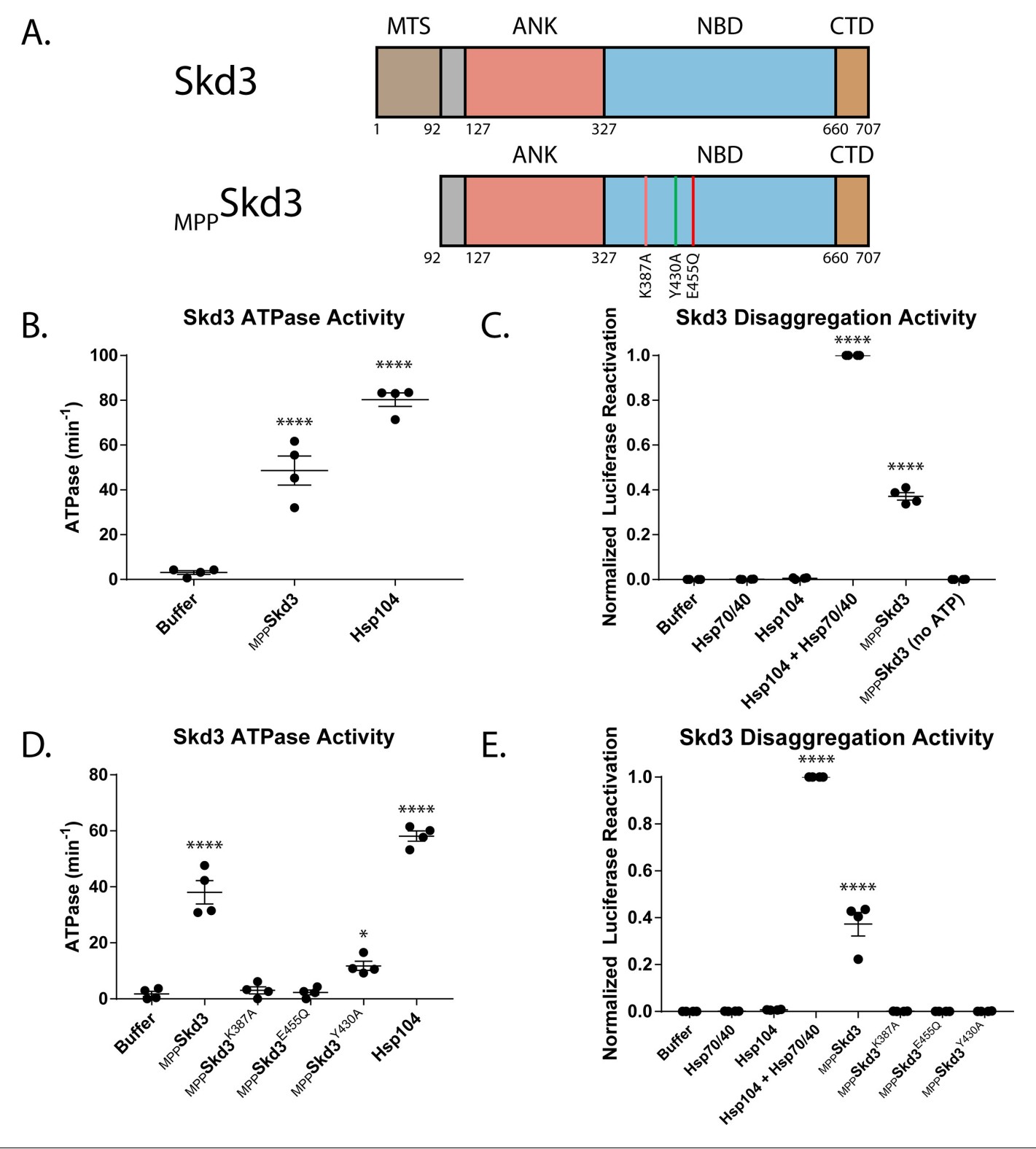

**Figure 2.** Skd3 is a protein disaggregase. (**A**) Domain map depicting the Mitochondrial-processing peptidase (MPP) cleavage site and mature-length Skd3 (MPPSkd3). The MTS was predicted using MitoProt in agreement with previous work, as outlined in the Materials and methods. The positions of the Walker A mutation (K387A) predicted to disrupt ATP binding and hydrolysis, pore-loop tyrosine mutation (Y430A) predicted to disrupt substrate binding, and Walker B mutation (E455Q) predicted to disrupt ATP hydrolysis are shown. (**B**) MPPSkd3 is an ATPase. ATPase assay comparing MPPSkd3 and Hsp104. MPPSkd3 and Hsp104 ATPase were compared to buffer using one-way ANOVA and a Dunnett's multiple comparisons test (N = 4,

*Figure 2 continued on next page*

Figure 2 continued

individual data points shown as dots, bars show mean ± SEM, ****p<0.0001). (C) Luciferase disaggregation/reactivation assay showing that <sub>MPP</sub>Skd3 has disaggregase activity in the presence but not absence of ATP. Luciferase activity was buffer subtracted and normalized to Hsp104 + Hsp70/Hsp40. Luciferase activity was compared to buffer using one-way ANOVA and a Dunnett's multiple comparisons test (N = 4, individual data points shown as dots, bars show mean ± SEM, ****p<0.0001). (D) ATPase assay comparing <sub>MPP</sub>Skd3, <sub>MPP</sub>Skd3$^{K387A}$ (Walker A mutant), <sub>MPP</sub>Skd3$^{E455Q}$ (Walker B mutant), and <sub>MPP</sub>Skd3$^{Y430A}$ (Pore-Loop mutant), showing that both Walker A and Walker B mutations abolish Skd3 ATPase activity, whereas the Pore Loop mutation reduces ATPase activity. ATPase activity was compared to buffer using one-way ANOVA and a Dunnett's multiple comparisons test (N = 4, individual data points shown as dots, bars show mean ± SEM, *p<0.05, ****p<0.0001). (E) Luciferase disaggregation/reactivation assay comparing <sub>MPP</sub>Skd3 to Walker A, Walker B, and Pore-Loop variants demonstrating that ATP binding, ATP hydrolysis, and pore-loop contacts are essential for Skd3 disaggregase activity. Luciferase activity was buffer subtracted and normalized to Hsp104 + Hsp70/Hsp40. Luciferase activity was compared to buffer using one-way ANOVA and a Dunnett's multiple comparisons test (N = 4, individual data points shown as dots, bars show mean ± SEM, ****p<0.0001). The online version of this article includes the following figure supplement(s) for figure 2:

**Figure supplement 1.** Recombinant Skd3 is highly pure and immunoreactive with several commercially available antibodies.
**Figure supplement 2.** Skd3 is a protein disaggregase.

We next investigated the role of conserved AAA+ elements in Skd3 activity. Thus, we mutated: (1) a critical lysine in the Walker A motif to alanine (K387A), which is predicted to disrupt ATP binding and hydrolysis (*Hanson and Whiteheart, 2005*; *Seraphim and Houry, 2020*); (2) a critical glutamate in the Walker B motif to glutamine (E455Q), which is predicted to disrupt ATP hydrolysis but not ATP binding (*Hanson and Whiteheart, 2005*; *Seraphim and Houry, 2020*); and (3) a highly-conserved tyrosine in the predicted -GYVG- substrate-binding loop to alanine that is predicted to disrupt substrate binding (Y430A) as in related HCLR clade AAA+ ATPases (*Gates et al., 2017*; *Hanson and Whiteheart, 2005*; *Lopez et al., 2020*; *Martin et al., 2008*; *Rizo et al., 2019*; *Seraphim and Houry, 2020*). The equivalent Walker A, Walker B, and substrate-binding loop mutations in NBD1 and NBD2 of Hsp104 severely disrupt disaggregase activity (*DeSantis et al., 2012*; *Lum et al., 2004*; *Torrente et al., 2016*). Likewise, <sub>MPP</sub>Skd3$^{K387A}$ (Walker A mutant) and <sub>MPP</sub>Skd3$^{E455Q}$ (Walker B mutant) displayed greatly reduced ATPase and disaggregase activity (*Figure 2d,e*). Thus, <sub>MPP</sub>Skd3 couples ATP binding and hydrolysis to protein disaggregation.

Interestingly, the pore-loop variant, <sub>MPP</sub>Skd3$^{Y430A}$, exhibited reduced ATPase activity compared to <sub>MPP</sub>Skd3, but much higher ATPase activity than <sub>MPP</sub>Skd3$^{K387A}$ and <sub>MPP</sub>Skd3$^{E455Q}$ (*Figure 2d*). This reduction in ATPase activity was unexpected as equivalent mutations in Hsp104 do not affect ATPase activity (*DeSantis et al., 2012*; *Lum et al., 2008*; *Lum et al., 2004*; *Torrente et al., 2016*). <sub>MPP</sub>Skd3$^{Y430A}$ was also devoid of disaggregase activity (*Figure 2e*). The inhibition of disaggregase activity by Y430A was much more severe than the inhibition of ATPase activity (*Figure 2d,e*), which suggests that the pore-loop Y430 might make direct contact with substrate to drive protein disaggregation as in Hsp104 (*DeSantis et al., 2012*; *Gates et al., 2017*). Thus, the conserved putative substrate-binding tyrosine of the -GYVG- pore-loop is critical for <sub>MPP</sub>Skd3 disaggregase activity.

## Skd3 disaggregase activity is enhanced by PARL cleavage

We noticed that Skd3 contains an undefined, 35-amino acid, hydrophobic stretch between the N-terminal MTS and the ankyrin-repeat domain (*Figure 1a* and *Figure 3—figure supplement 1*). Intriguingly, Skd3 is cleaved by the inner-membrane rhomboid protease, PARL, at amino acid 127, between the 35-amino acid, hydrophobic stretch and the ankyrin-repeat domain (*Figure 3—figure supplement 1*; *Saita et al., 2017*; *Spinazzi et al., 2019*). Sequence analysis shows that the 35-amino acid, hydrophobic stretch and the PARL-cleavage motif are both highly conserved among mammalian Skd3 orthologues (*Figure 3—figure supplement 2a*). Thus, we hypothesized that this 35-amino acid, hydrophobic stretch might be auto-inhibitory for Skd3 activity.

To determine whether PARL cleavage of this 35-amino acid, hydrophobic stretch regulates Skd3 activity, we purified Skd3 without this region (<sub>PARL</sub>Skd3) (*Figure 3a*). We found that the absence of the 35-residue stretch that is removed by PARL slightly decreased Skd3 ATPase activity compared to <sub>MPP</sub>Skd3 (*Figure 3b* and *Figure 3—figure supplement 2b*). Moreover, we find that <sub>MPP</sub>Skd3 ATPase activity is not stimulated by the model substrate, casein, a classic peptide-stimulator of Hsp104 ATPase activity (*Figure 3—figure supplement 3*; *Cashikar et al., 2002*; *Gates et al., 2017*). By contrast, <sub>PARL</sub>Skd3 ATPase activity is mildly stimulated by casein (*Figure 3—figure supplement*

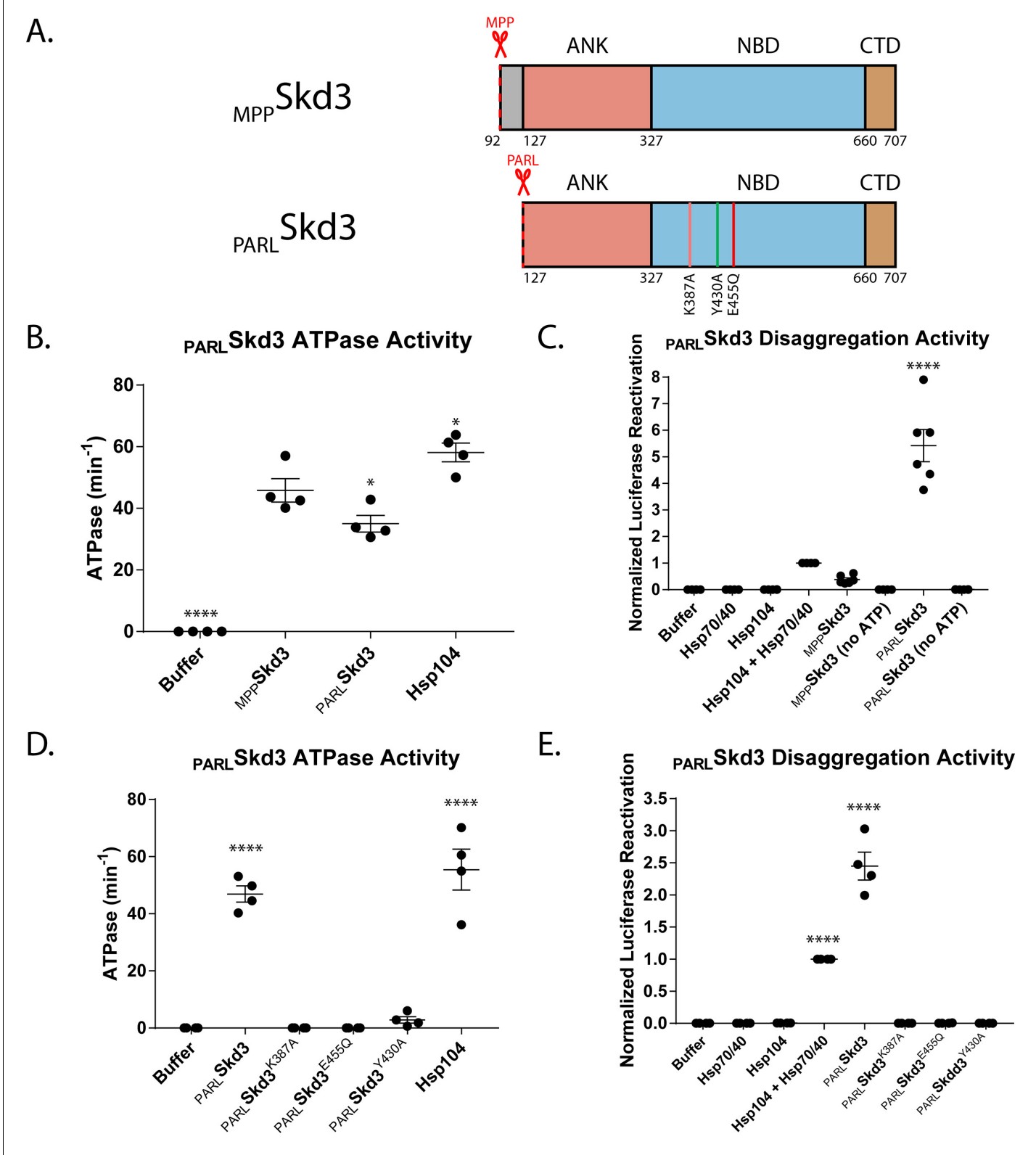

**Figure 3.** PARL cleavage enhances Skd3 disaggregase activity. (**A**) Domain map depicting $_{MPP}$Skd3 and the PARL cleavage site and corresponding PARL-cleaved Skd3 ($_{PARL}$Skd3). The positions of the Walker A mutation (K387A) predicted to disrupt ATP binding and hydrolysis, pore-loop tyrosine mutation (Y430A) predicted to disrupt substrate binding, and Walker B mutation (E455Q) predicted to disrupt ATP hydrolysis are shown. (**B**) ATPase assay comparing $_{MPP}$Skd3 and $_{PARL}$Skd3. $_{PARL}$Skd3 is catalytically active, but is slightly less active than $_{MPP}$Skd3. $_{PARL}$Skd3 and Hsp104 ATPase were

*Figure 3 continued on next page*

Figure 3 continued

compared to $_{MPP}$Skd3 ATPase using one-way ANOVA and a Dunnett's multiple comparisons test (N = 6, individual data points shown as dots, bars show mean ± SEM, *p<0.05, ****p<0.0001). (C) Luciferase disaggregation/reactivation assay comparing $_{MPP}$Skd3 disaggregase activity to $_{PARL}$Skd3. $_{PARL}$Skd3 was over 10-fold more active than $_{MPP}$Skd3. Luciferase activity was buffer subtracted and normalized to Hsp104 + Hsp70/Hsp40. Luciferase activity was compared to $_{MPP}$Skd3 using one-way ANOVA and a Dunnett's multiple comparisons test (N = 6, individual data points shown as dots, bars show mean ± SEM, ****p<0.0001). (D) ATPase assay comparing $_{PARL}$Skd3, $_{PARL}$Skd3$^{K387A}$ (Walker A), $_{PARL}$Skd3$^{E455Q}$ (Walker B), and $_{PARL}$Skd3$^{Y430A}$ (Pore Loop), showing that both Walker A and Walker B mutations abolish Skd3 ATPase activity, whereas the Pore-Loop mutation reduces ATPase activity. ATPase activity was compared to buffer using one-way ANOVA and a Dunnett's multiple comparisons test (N = 4, individual data points shown as dots, bars show mean ± SEM, ****p<0.0001). (E) Luciferase disaggregation/reactivation assay comparing $_{PARL}$Skd3 to Walker A, Walker B, and Pore-Loop variants showing that ATP binding, ATP hydrolysis, and pore-loop contacts are essential for $_{PARL}$Skd3 disaggregase activity. Luciferase activity was buffer subtracted and normalized to Hsp104 + Hsp70/Hsp40. Luciferase activity was compared to buffer using one-way ANOVA and a Dunnett's multiple comparisons test (N = 4, individual data points shown as dots, bars show mean ± SEM, ****p<0.0001).

The online version of this article includes the following figure supplement(s) for figure 3:

**Figure supplement 1.** The auto-inhibitory peptide of Skd3 is hydrophobic.
**Figure supplement 2.** PARL cleavage of Skd3 enhances Skd3 disaggregase activity.
**Figure supplement 3.** $_{PARL}$Skd3, but not $_{MPP}$Skd3, ATPase activity is stimulated by a model substrate.

3). This finding indicates that $_{PARL}$Skd3 may interact more effectively with substrates than $_{MPP}$Skd3 due to the removal of the 35-amino acid, hydrophobic stretch.

Remarkably, the absence of the N-terminal peptide known to be removed by PARL unleashed Skd3 disaggregase activity (*Figure 3c* and *Figure 3—figure supplement 2c,d*). Indeed, $_{PARL}$Skd3 exhibited over 10-fold higher disaggregase activity (~44.06 ± 3.78% native luciferase activity recovered, mean ± SEM) compared to $_{MPP}$Skd3 (~3.10 ± 0.44% native luciferase activity recovered) and over five-fold higher disaggregation activity than Hsp104 plus Hsp70 and Hsp40 (~8.58 ± 1.20% native luciferase activity recovered), despite $_{PARL}$Skd3 having lower ATPase activity when compared to $_{MPP}$Skd3 (*Figure 3b,c*, and *Figure 3—figure supplement 2b,c,d*). These results suggest that Skd3 disaggregase activity is likely regulated by PARL and that PARL-activated Skd3 is a powerful, stand-alone protein disaggregase with activity comparable to potentiated Hsp104 variants (*Jackrel et al., 2014*; *Jackrel and Shorter, 2014*; *Jackrel et al., 2015*; *Tariq et al., 2019*; *Tariq et al., 2018*; *Torrente et al., 2016*).

As with $_{MPP}$Skd3, we found that $_{PARL}$Skd3 disaggregase activity was supported by ATP, but not in the absence of nucleotide or in the presence of ADP, non-hydrolyzable AMP-PNP, or slowly hydrolyzable ATPγS (*Figure 3c* and *Figure 3—figure supplement 2e*). Likewise, $_{PARL}$Skd3$^{K387A}$ (Walker A mutant) and $_{PARL}$Skd3$^{E455Q}$ (Walker B mutant) lacked ATPase and disaggregase activity (*Figure 3d, e*), indicating that $_{PARL}$Skd3 couples ATP binding and hydrolysis to protein disaggregation. Curiously, $_{PARL}$Skd3$^{Y430A}$ (pore-loop mutant) exhibited a larger reduction in ATPase activity than $_{MPP}$Skd3$^{Y430A}$ (*Figures 2d* and *3d*), indicating that the conserved tyrosine in the conserved putative substrate-binding -GYVG- pore loop impacts ATPase activity in Skd3, whereas it has no effect in Hsp104 (*DeSantis et al., 2012*; *Torrente et al., 2016*). $_{PARL}$Skd3$^{Y430A}$ was devoid of disaggregase activity (*Figure 3e*), which could be due to reduced ATPase activity, reduced substrate binding, or both.

## PARL-activated Skd3 solubilizes α-synuclein fibrils

Next, we assessed whether $_{PARL}$Skd3 could disassemble a stable amyloid substrate, which makes more stringent demands on a disaggregase (*DeSantis et al., 2012*). Thus, we turned to α-synuclein fibrils, which are connected to Parkinson's disease and various synucleinopathies (*Henderson et al., 2019*; *Spillantini et al., 1998a*; *Spillantini et al., 1998b*). We utilized a strain of synthetic α-synuclein fibrils capable of eliciting Parkinson's disease-like symptoms in mice (*Luk et al., 2012*). We employed samples of preformed α-synuclein fibrils that had completely assembled (i.e. 100% of α-synuclein was in the assembled fibril state) and were devoid of soluble α-synuclein (*Figure 4a,b*). Using a sedimentation assay combined with a dot-blot, we found that $_{PARL}$Skd3, but not $_{MPP}$Skd3 (data not shown), disaggregated these disease-causing fibrils in the presence, but not absence of ATP (*Figure 4a,b*). Thus, $_{PARL}$Skd3 is a powerful protein disaggregase, which could have applications as a potential therapeutic agent to eliminate disease-causing α-synuclein fibrils that underlie Parkinson's disease and other synucleinopathies.

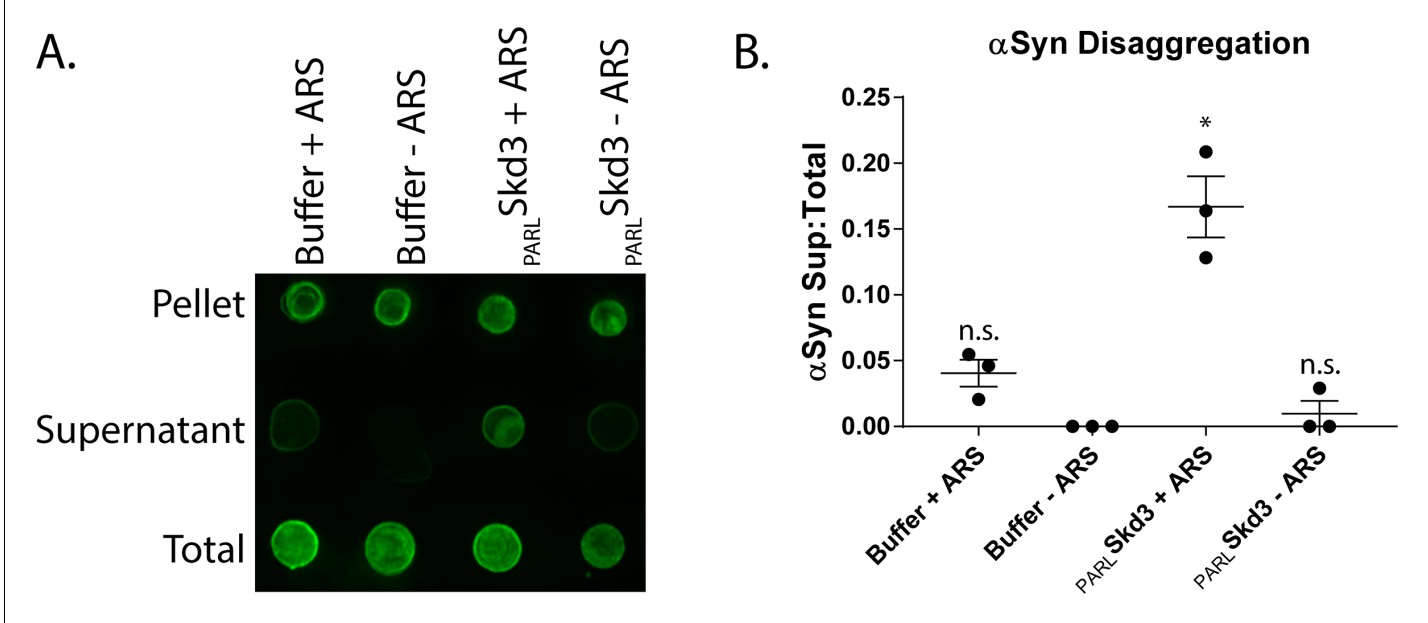

**Figure 4.** Skd3 disaggregates α-synuclein fibrils. (**A**) Representative dot blot of α-synuclein disaggregation assay. Blot shows solubilization of α-synuclein fibrils by ₚₐᵣₗSkd3 in the presence of an ATP regeneration system (ARS), but not in the presence of ₚₐᵣₗSkd3 or ARS alone. (N = 3). (**B**) Quantification of α-synuclein disaggregation assay showing that ₚₐᵣₗSkd3 in the presence of an ARS disaggregates α-synuclein fibrils. Results are normalized as fraction in the supernatant relative to the fraction in the supernatant and the pellet. The fraction of α-synuclein in the supernatant was compared to buffer using a repeated measure one-way ANOVA and a Dunnett's multiple comparisons test (N = 3, individual data points shown as dots, bars show mean ± SEM, *p<0.05).

## Skd3 disaggregase activity requires the ankyrin-repeat domain and NBD

To further investigate the mechanism of Skd3 disaggregase activity, we purified the isolated ankyrin-repeat domain and NBD as separate proteins (*Figure 5a*). Neither the isolated ankyrin-repeat domain nor the isolated NBD exhibited ATPase activity or disaggregase activity (*Figure 5b,c*). The lack of ATPase activity and disaggregase activity of the isolated NBD is consistent with a similar lack of activity of isolated NBD2 from Hsp104 or bacterial ClpB and is likely attributed to a lack of hexamer formation, as is observed for ClpB NBD2 (*Beinker et al., 2005*; *Hattendorf and Lindquist, 2002*; *Mogk et al., 2003*). Thus, the ankyrin-repeat domain and NBD combine in *cis* to enable Skd3 ATPase activity and disaggregase activity. We also tested whether the two domains could combine in *trans* as two separate proteins to yield an active ATPase or disaggregase. However, we found that equimolar amounts of the ankyrin-repeat domain and NBD were also inactive (*Figure 5b,c*). Thus, Skd3 is unlike bacterial ClpB, which can be reconstituted in *trans* by separate NTD-NBD1-MD and NBD2 proteins (*Beinker et al., 2005*). These findings suggest that the covalent linkage of the ankyrin-repeat domain and NBD is critical for forming a functional ATPase and disaggregase. Importantly, they also predict that truncated MGCA7-linked Skd3 variants, such as R250* and K321* (where * indicates a stop codon), which lack the NBD would be inactive for protein disaggregation and indeed any ATPase-dependent activities.

## Skd3 disaggregase activity is not stimulated by Hsp70 and Hsp40

Hsp104 and Hsp78 collaborate with Hsp70 and Hsp40 to disaggregate many substrates (*Figures 2c, e* and *3c,e*; *DeSantis et al., 2012*; *Glover and Lindquist, 1998*; *Krzewska et al., 2001*). By contrast, Skd3 does not require Hsp70 and Hsp40 for protein disaggregation (*Figures 2c,e*, *3c,e* and *4a, b*). This finding is consistent with Skd3 lacking the NTD, NBD1, and MD of Hsp104, which interact with Hsp70 (*DeSantis et al., 2014*; *Lee et al., 2013*; *Sweeny et al., 2015*; *Sweeny et al., 2020*). Nonetheless, Hsp70 and Hsp40 might still augment Skd3 disaggregase activity. Thus, we tested if Hsp70 and Hsp40 could stimulate Skd3 disaggregase activity. However, neither ₘₚₚSkd3 nor

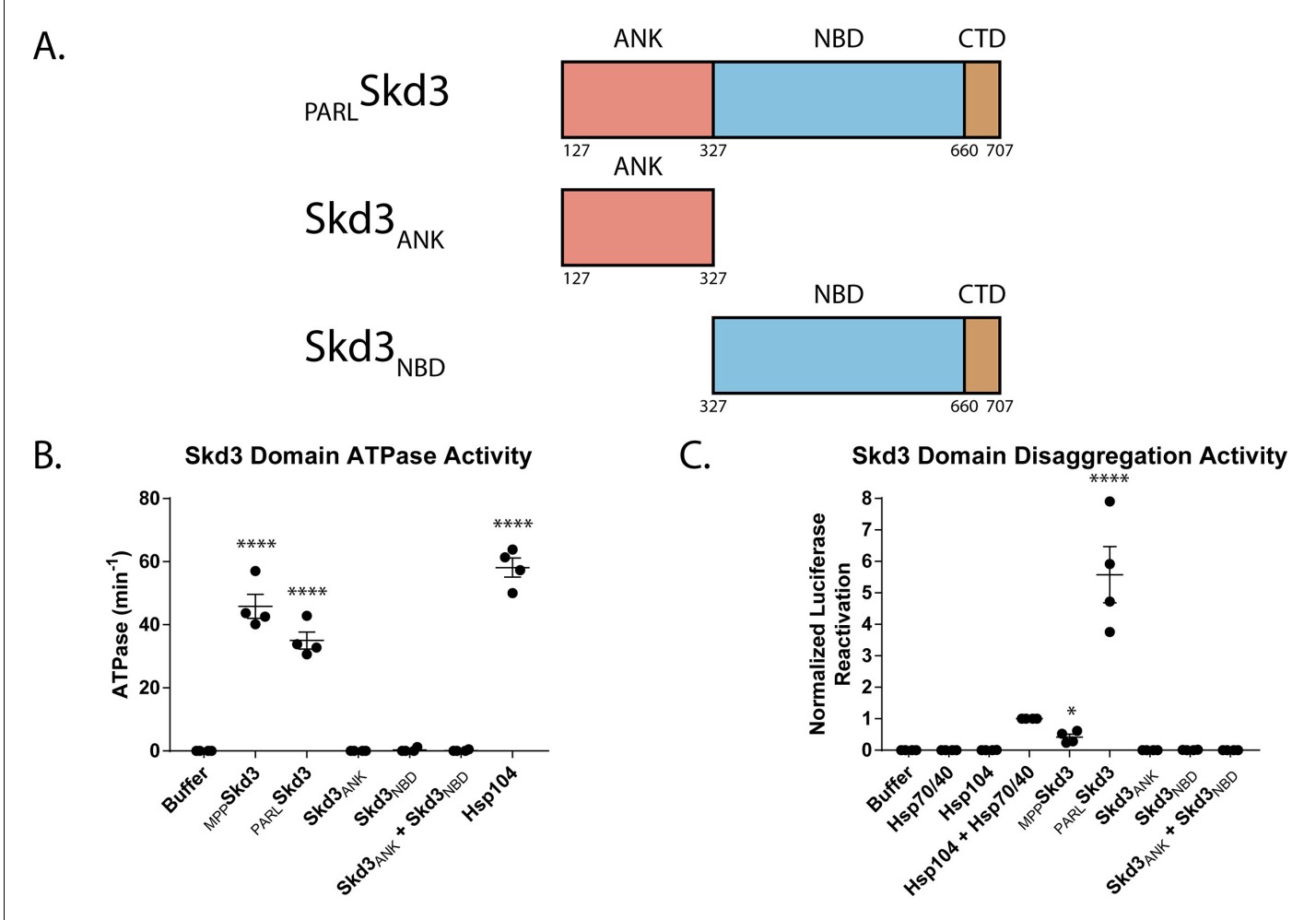

**Figure 5.** The ankyrin-repeat domain and NBD are required for Skd3 disaggregase activity. (**A**) Domain maps showing the Skd3$_{ANK}$ and Skd3$_{NBD}$ constructs. (**B**) ATPase assay comparing Skd3$_{ANK}$ and Skd3$_{NBD}$ ATPase activity. Results show that Skd3$_{ANK}$, Skd3$_{NBD}$, and Skd3$_{ANK}$ + Skd3$_{NBD}$ do not have ATPase activity. Data are from the same experiments as *Figure 3B*. ATPase activity was compared to buffer using one-way ANOVA and a Dunnett's multiple comparisons test (N = 4, individual data points shown as dots, bars show mean ± SEM, ****p<0.0001). (**C**) Luciferase disaggregation/reactivation assay comparing Skd3$_{ANK}$, Skd3$_{NBD}$, and Skd3$_{ANK}$ + Skd3$_{NBD}$ d activity. Results show that Skd3$_{ANK}$, Skd3$_{NBD}$, or Skd3$_{ANK}$ + Skd3$_{NBD}$ are inactive disaggregases. Data are from same experiments as *Figure 3C*. Luciferase activity was buffer subtracted and normalized to Hsp104 plus Hsp70 and Hsp40. Luciferase disaggregase activity was compared to buffer using one-way ANOVA and a Dunnett's multiple comparisons test (N = 4, individual data points shown as dots, bars show mean ± SEM, ****p<0.0001).

$_{PARL}$Skd3disaggregase activity was stimulated by Hsp70 and Hsp40 (*Figure 6a,b*). Thus, Skd3 is a stand-alone disaggregase that works independently of the Hsp70 chaperone system.

## Human cells lacking Skd3 exhibit reduced solubility of mitochondrial proteins

Given the potent disaggregase activity of Skd3, we predicted that deletion of Skd3 in human cells would result in decreased protein solubility in mitochondria. To determine the effect of Skd3 on protein solubility in mitochondria, we compared the relative solubility of the mitochondrial proteome in wild-type and Skd3 knockout human HAP1 cells (*Figure 7—figure supplement 1a–c*) using mass spectrometry as described in *Figure 7a*. Overall, we observed decreased protein solubility in mitochondria from the Skd3 knockout cells when compared to their wild-type counterparts (*Figure 7b* and *Figure 7—figure supplement 2a*). Using Gene Ontology (GO) analysis for cellular component, we found that proteins in the inner mitochondrial membrane and intermembrane space were

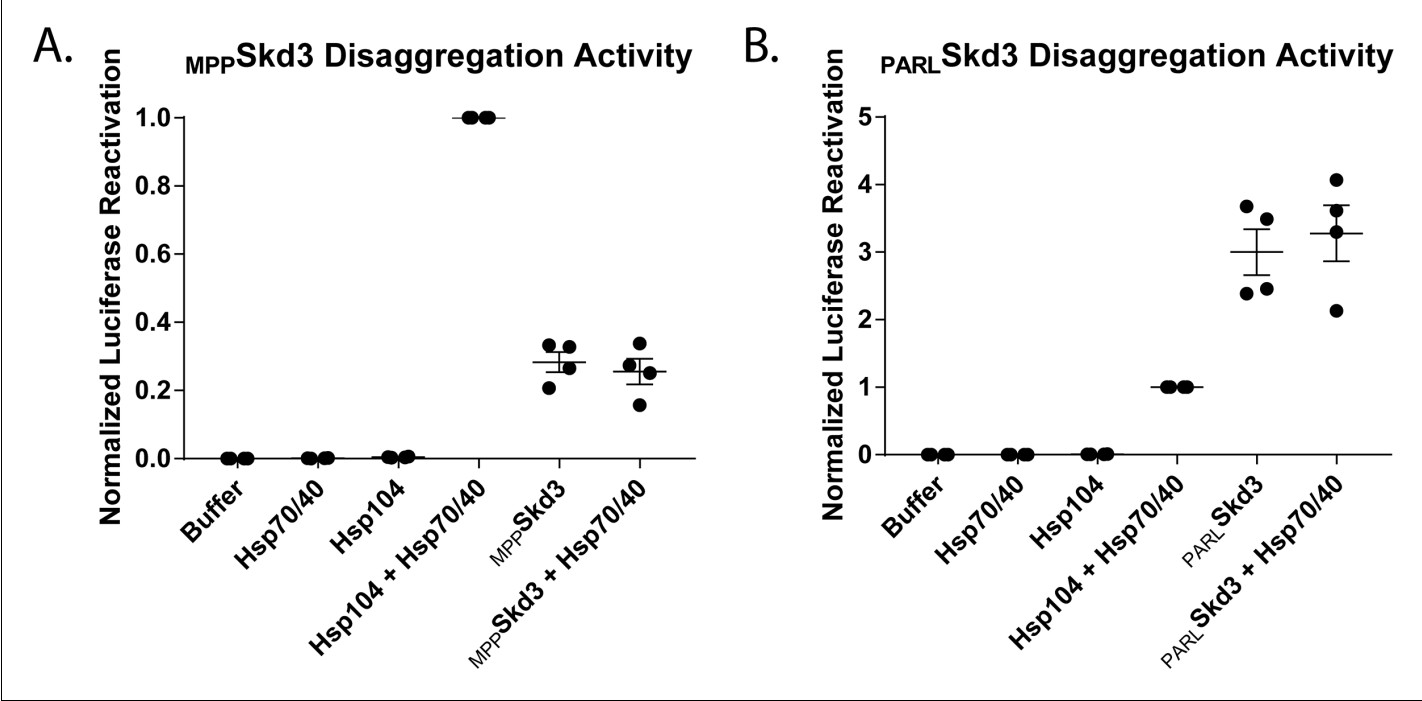

**Figure 6.** Skd3 does not collaborate with Hsp70 and Hsp40 in protein disaggregation. (**A**) Luciferase disaggregation/reactivation comparing $_{MPP}$Skd3 disaggregase activity in the presence or absence of Hsp70 (Hsc70) and Hsp40 (Hdj1). Luciferase activity was buffer subtracted and normalized to Hsp104 plus Hsp70 and Hsp40. Results show a stimulation of Hsp104 disaggregase activity by Hsp70 and Hsp40, but no stimulation of disaggregase activity for $_{MPP}$Skd3. $_{MPP}$Skd3 plus Hsp70 and Hsp40 was compared to $_{MPP}$Skd3 using a two-tailed, unpaired t-test. Test found no significant difference in disaggregation activity. (N = 4, individual data points shown as dots, bars show mean ± SEM). (**B**) Luciferase disaggregation/reactivation comparing $_{PARL}$Skd3 disaggregase activity in the presence or absence of Hsp70 and Hsp40. Luciferase activity was buffer subtracted and normalized to Hsp104 plus Hsp70 and Hsp40. Results show no stimulation of disaggregase activity for $_{PARL}$Skd3 by Hsp70 and Hsp40. $_{PARL}$Skd3 plus Hsp70 and Hsp40 was compared to $_{PARL}$Skd3 using a two-tailed, unpaired t-test. Test found no significant difference in disaggregation activity. (N = 4, individual data points shown as dots, bars show mean ± SEM).

enriched in the insoluble fraction in the absence of Skd3 (*Figure 7—figure supplement 2a*; *Ashburner et al., 2000*; *Mi et al., 2019*; *The Gene Ontology Consortium, 2019*). Importantly, Skd3 has been localized to the mitochondrial intermembrane space (*Botham et al., 2019*; *Hung et al., 2014*; *Yoshinaka et al., 2019*). Using GO analysis for biological process, we found that proteins involved in calcium import into mitochondria, chaperone-mediated protein transport, protein insertion into the mitochondrial inner membrane, mitochondrial electron transport, mitochondrial respiratory-chain complex assembly, and cellular response to hypoxia are more insoluble in Skd3 knockout cells compared to wild-type cells (*Figure 7c* and *Figure 7—figure supplement 2b*; *Ashburner et al., 2000*; *Mi et al., 2019*; *The Gene Ontology Consortium, 2019*).

Specifically, we find that HAX1, OPA1, PHB2, PARL, SMAC/DIABLO, and HTRA2 are more insoluble, which implicates a key role for Skd3 in regulating apoptotic and proteolytic pathways (*Baumann et al., 2014*; *Chai et al., 2000*; *Chao et al., 2008*; *Cipolat et al., 2006*; *Frezza et al., 2006*; *Klein et al., 2007*; *Saita et al., 2017*; *Yoshinaka et al., 2019*). Additionally, the regulators of the mitochondrial calcium uniporter (MCU), MICU1 and MICU2 along with several members of the SLC25 family (including the calcium binding SLC25A25 and SLC25A13) were found to be more insoluble in the knockout compared to the wild type, implicating Skd3 in the regulation of mitochondrial calcium transport and signaling (*Anunciado-Koza et al., 2011*; *Csordás et al., 2013*; *Palmieri et al., 2001*; *Patron et al., 2014*; *Perocchi et al., 2010*; *Plovanich et al., 2013*). Indeed, a sedimentation assay as described in *Figure 7a* in combination with western blot analysis shows an overall decrease in solubility of MICU2 in mitochondria lacking Skd3 (*Figure 7—figure supplement 3a,b*). We also observed an enrichment of translocase of the inner membrane (TIM) components, TIMM8A, TIMM8B, TIMM13, TIMM21, TIMM22, TIMM23, and TIMM50 in the insoluble fraction of Skd3

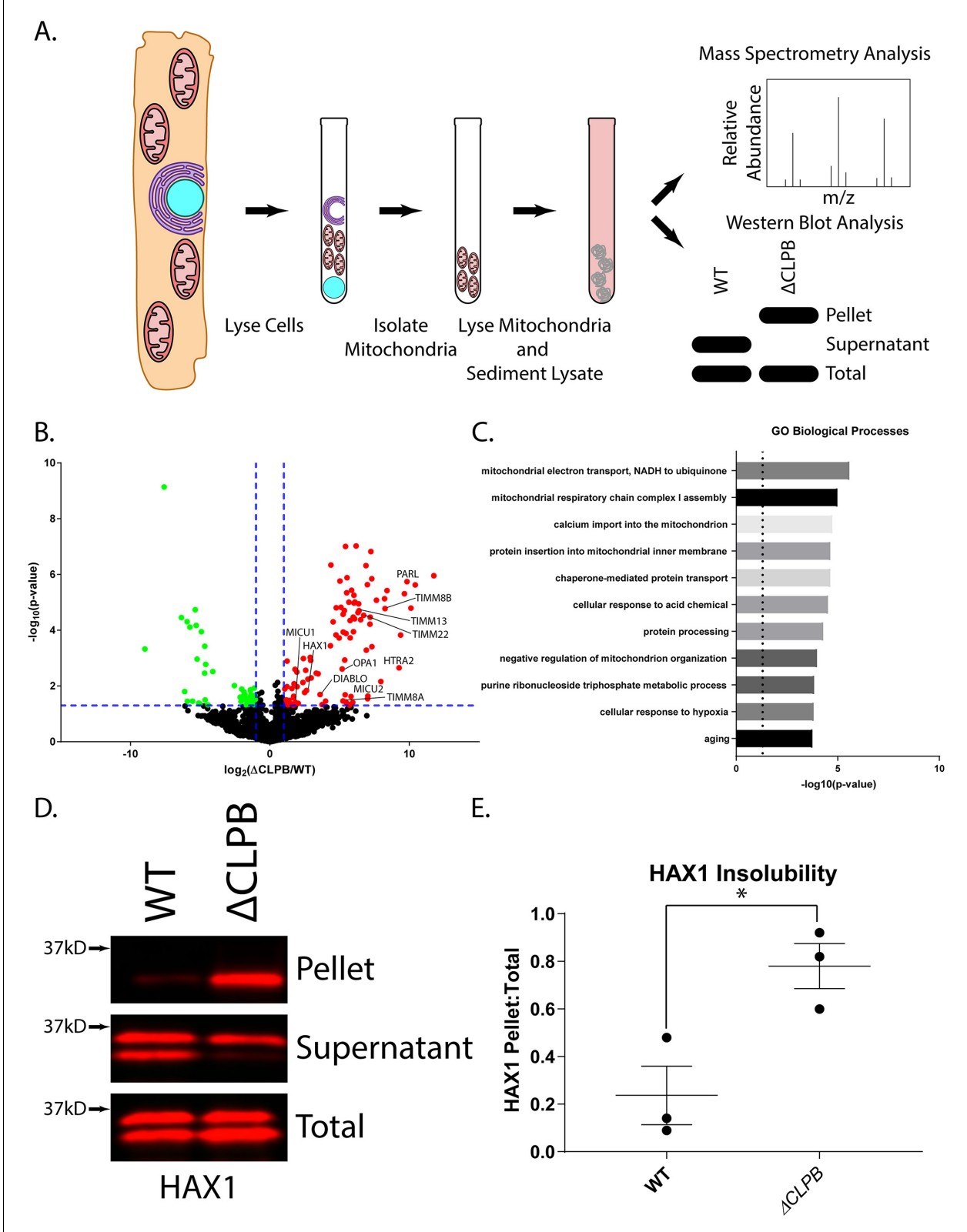

**Figure 7.** Skd3 maintains the solubility of key mitochondrial proteins in human cells. (**A**) Schematic showing sedimentation assay design. HAP1 cells were lysed and the mitochondrial fraction was separated from the cytosolic fraction. The mitochondrial fraction was then lysed, and the soluble fraction was separated from the insoluble fraction via sedimentation. The samples were then either analyzed via mass-spectrometry or western blotting. (**B**) Volcano plot showing the $\log_2$ fold change of protein in the Skd3 (*CLPB*) knockout pellet compared to the wild-type (WT) pellet. The 99 proteins that

*Figure 7 continued on next page*

*Figure 7 continued*

were enriched in the Skd3 pellet are highlighted in red. The 53 proteins that were enriched in the wild-type (WT) pellet are highlighted in green. Significance cutoffs were set as fold change >2.0 and p<0.05, indicated with blue dashed lines (N = 3, p<0.05). (**C**) Select statistically significant terms for GO biological processes from the enriched proteins in the Skd3 knockout pellet. Dashed line shows p=0.05 (p<0.05). For full list see *Figure 7— figure supplement 2b*. (**D**) Representative western blot of sedimentation assay showing relative solubility of HAX1 protein in wild-type (WT) and Skd3 (*CLPB*) knockout cells. Results show a marked decrease in HAX1 solubility when Skd3 is knocked out. (N = 3). (**E**) Quantification of HAX1 sedimentation assay shows an overall increase in the insoluble HAX1 relative to the total protein in the Skd3 (*CLPB*) knockout cell line. Quantification is normalized as signal in the pellet divided by the sum of the signal in the pellet and supernatant. The fraction in the pellet for the Skd3 knockout was compared to the wild-type cells using a two-way, unpaired, t-test. (N = 3, individual data points shown as dots, bars show mean ± SEM, *p<0.05).

The online version of this article includes the following figure supplement(s) for figure 7:

**Figure supplement 1.** Verification of Skd3 knockout and mitochondria isolation in HAP1 cells.
**Figure supplement 2.** Skd3 deletion increases insolubility of mitochondrial inner membrane and intermembrane space proteins.
**Figure supplement 3.** Skd3 maintains the solubility of MICU2 in human cells.
**Figure supplement 4.** HAX1 is a highly disordered protein.

knockouts (*Chacinska et al., 2005*; *Donzeau et al., 2000*; *Geissler et al., 2002*; *Meinecke et al., 2006*; *Mokranjac et al., 2003*; *Paschen et al., 2000*; *Sirrenberg et al., 1996*; *Yamamoto et al., 2002*). Finally, we observed an enrichment in respiratory complex I and III proteins and their assembly factors such as NDUFA8, NDUFA11, NDUFA13, NDUFB7, NDUFB10, TTC19, COX11, and CYC1 in the insoluble fraction of Skd3 knockouts (*Figure 6b* and *Supplementary file 2*; *Angebault et al., 2015*; *Ghezzi et al., 2011*; *Spinazzi et al., 2019*; *Szklarczyk et al., 2011*; *Tzagoloff et al., 1990*). Overall, these results suggest an important role for Skd3 in maintaining the solubility of proteins of the inner mitochondrial membrane and intermembrane space, including key regulators in apoptosis, mitochondrial calcium regulation, protein import, and respiration. Thus, in cells Skd3 appears critical for protein solubility in the intermembrane space and mitochondrial inner membrane.

## Skd3 promotes HAX1 solubility in human cells

HAX1 is a highly-disordered protein that has been previously identified as a Skd3 substrate both in cells and in silico (*Figure 7—figure supplement 4*; *Chen et al., 2019*; *Wortmann et al., 2015*). HAX1 is an anti-apoptotic BCL-2 family protein that enables efficient cleavage of HTRA2 by PARL to promote cell survival (*Chao et al., 2008*; *Klein et al., 2007*). To test if Skd3 regulates HAX1 solubility in human cells, we compared the solubility of HAX1 in wild-type and Skd3-knockout HAP1 cells via sedimentation analysis and western blot. In wild-type cells, HAX1 remained predominantly soluble (*Figure 7d,e*). However, when Skd3 was deleted HAX1 became predominantly insoluble (*Figure 7d, e*). Thus, Skd3 is essential for HAX1 solubility in cells. Curiously, loss of Skd3 has been previously shown to promote apoptosis in specific contexts (*Chen et al., 2019*). Furthermore, HAX1 stability has been implicated as a regulator of apoptotic signaling (*Baumann et al., 2014*). Our data support a model whereby Skd3 exerts its anti-apoptotic effect by maintaining HAX1 solubility and contingent functionality.

## MGCA7-linked Skd3 variants display diminished disaggregase activity

Skd3 has been implicated in a severe mitochondrial disorder, MGCA7, yet little is known about its contribution or function in this disease (*Capo-Chichi et al., 2015*; *Kanabus et al., 2015*; *Kiykim et al., 2016*; *Pronicka et al., 2017*; *Saunders et al., 2015*; *Wortmann et al., 2016*; *Wortmann et al., 2015*). Indeed, many mutations in Skd3 are connected with MGCA7 (*Figure 8a*). Most of these are clustered in the NBD, but several are also in the ankyrin-repeat domain, and one frameshift is found in the mitochondrial targeting signal (*Figure 8a*). Some MGCA7-linked Skd3 variants, such as R250* and K321* (where * indicates a stop codon), lack the NBD and would be predicted in light of our findings to be incapable protein disaggregation and indeed any ATPase-dependent activities (*Figure 5b,c*). We hypothesized that MGCA7-linked missense mutations also directly affect Skd3 disaggregase activity. To test this hypothesis, we purified MGCA7-linked variants of Skd3 from cases where both patient alleles bear the mutation, specifically: T268M, R475Q, A591V, and R650P (*Pronicka et al., 2017*). These Skd3 variants cause MGCA7 on a spectrum of clinical severity, which is defined by a scoring system assigned by physicians based on the presentation of various symptoms associated with MGCA7 (*Pronicka et al., 2017*). Broadly, the severe phenotype

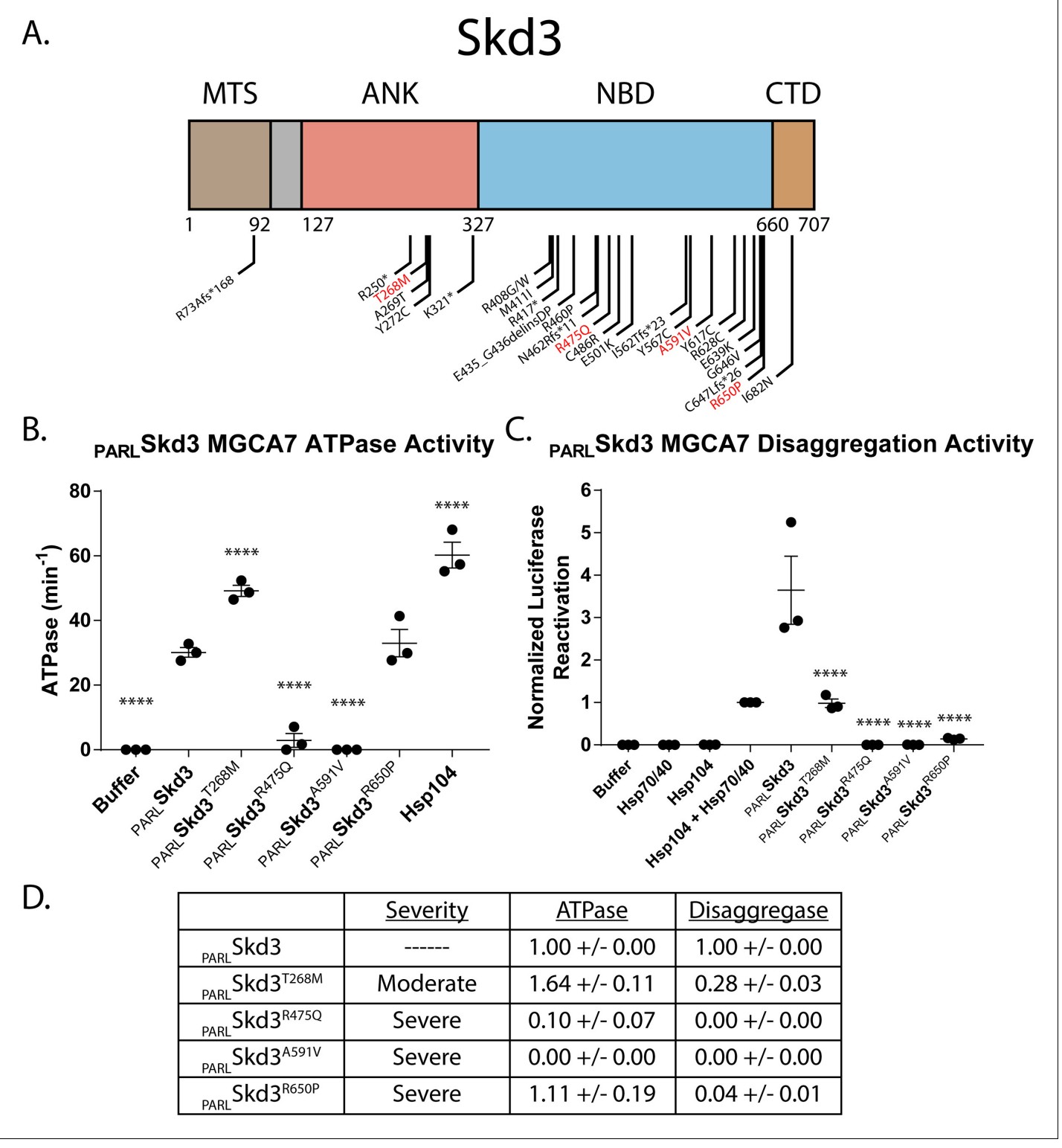

**Figure 8.** Skd3 disaggregase activity predicts the clinical severity of MGCA7-associated mutations. (**A**) Domain map depicting all published mutations in Skd3 that have been associated with MGCA7. Mutants in red are studied further here. (**B**) ATPase assay showing the effect of four homozygous MGCA7 mutations on Skd3 activity. $_{PARL}$Skd3$^{T268M}$ has increased ATPase activity, $_{PARL}$Skd3$^{R475Q}$ and $_{PARL}$Skd3$^{A591V}$ have decreased ATPase activity, and $_{PARL}$Skd3$^{R650P}$ has unchanged ATPase activity compared to wild type. $_{PARL}$Skd3 MGCA7 mutants ATPase activities were compared to $_{PARL}$Skd3 wild-type using one-way ANOVA and a Dunnett's multiple comparisons test (N = 3, individual data points shown as dots, bars show mean ± SEM, ****p<0.0001). (**C**) Luciferase disaggregation/reactivation assay showing the effect of the same four homozygous MGCA7 mutations on Skd3 activity. $_{PARL}$Skd3$^{T268M}$

*Figure 8 continued on next page*

*Figure 8 continued*

had reduced disaggregase activity, whereas $_{PARL}$Skd3$^{R475Q}$, $_{PARL}$Skd3$^{A591V}$, and $_{PARL}$Skd3$^{R650P}$were almost completely inactive compared to wild type. Luciferase activity was buffer subtracted and normalized to Hsp104 plus Hsp70 and Hsp40. Luciferase disaggregase activity was compared to $_{PARL}$Skd3 wild type using one-way ANOVA and a Dunnett's multiple comparisons test (N = 3, individual data points shown as dots, bars show mean ± SEM, ****p<0.0001). (D) Table summarizing the clinical severity of each MGCA7 mutation as well as the ATPase activity and luciferase disaggregase activity. The table shows a relationship between luciferase disaggregase activity and clinical severity, but no relationship between either the ATPase activity and clinical severity or ATPase and luciferase disaggregase activity. Values represent ATPase activity and luciferase disaggregase activity normalized to wild-type $_{PARL}$Skd3 activity. Values show mean ± SEM (N = 3).

often presented with the absence of voluntary movements in neonates, hyperexcitability, ventilator dependency, and early death (*Pronicka et al., 2017*). The ankyrin-repeat variant, T268M, is linked to moderate MGCA7, whereas the NBD variants (R475Q, A591V, and R650P) are linked to severe MGCA7 (*Pronicka et al., 2017*).

Surprisingly, the ATPase activity varied for each MGCA7-linked variant. T268M had significantly increased ATPase activity, R475Q and A591V had marked decreased ATPase activity, and R650P ATPase was indistinguishable from wild type (*Figure 8b*). These ATPase activities did not correlate with clinical severity (*Figure 8d*; *Pronicka et al., 2017*). Thus, Skd3 variant ATPase activity does not accurately predict MGCA7 severity, as the mutation associated with mild MGCA7 had elevated ATPase relative to wild type, whereas different mutations capable of causing severe MGCA7 could exhibit impaired or nearly wild-type ATPase activity.

To address the disconnect between ATPase activity and MGCA7 disease severity, we next tested the disaggregase activity of these MGCA7-linked variants. Strikingly, and in contrast to ATPase activity, we found disaggregase activity tracks closely with disease severity. T268M, the only moderate phenotype variant tested, had ~27% disaggregase activity compared to wild type. By contrast, the three severe MGCA7 variants, R475Q, A591V, and R650P abolish or diminish disaggregation activity with 0%, 0%, and ~4% disaggregation activity compared to wild type, respectively (*Figure 8c*). Thus, disaggregase activity but not ATPase activity, is tightly correlated with clinical severity of MGCA7-linked mutations (*Figure 8d*; *Pronicka et al., 2017*). It will be of interest to test further MGCA7-linked variants to determine whether this trend holds. Nevertheless, our findings suggest that defects in Skd3 protein-disaggregase activity (and not other ATPase-related functions) are the driving factor in MGCA7 and pivotal to human health.

## Discussion

At the evolutionary transition from protozoa to metazoa, the potent mitochondrial AAA+ protein disaggregase, Hsp78, was lost (*Erives and Fassler, 2015*). Thus, it has long remained unknown whether metazoan mitochondria disaggregate and reactivate aggregated proteins. Here, we establish that another AAA+ protein, Skd3, is a potent metazoan mitochondrial protein disaggregase. Skd3 is activated by a mitochondrial inner-membrane rhomboid protease, PARL (*Figure 9*). PARL removes a hydrophobic auto-inhibitory sequence from the N-terminal region of Skd3, which prior to cleavage may limit Skd3 interactions with substrate (*Figure 9*). In this way, Skd3 only becomes fully activated as a disaggregase once it reaches its final cellular destination. Thus, cells might avoid any potential problems that could arise from unchecked Skd3 activity in the wrong cellular location. Skd3 activation by PARL may underlie several potential physiological mechanisms whereby Skd3 is either activated by PARL in response to certain cellular stressors or Skd3 is no longer activated by PARL upon the onset of apoptotic signaling.

Skd3 couples ATP binding and hydrolysis to protein disaggregation. To do so, Skd3 utilizes conserved AAA+ motifs in its NBD, including the Walker A and B motifs to bind and hydrolyze ATP, as well as a conserved pore-loop tyrosine, which likely engages substrate directly in a manner similar to other HCLR clade AAA+ proteins (*Erzberger and Berger, 2006*; *Gates et al., 2017*; *Lopez et al., 2020*; *Martin et al., 2008*; *Rizo et al., 2019*; *Seraphim and Houry, 2020*). However, the isolated NBD is insufficient for disaggregase activity, which indicates an important role for the ankyrin-repeat domain. Intriguingly, an ankyrin-repeat domain is also important for the activity of an unrelated ATP-independent protein disaggregase, cpSRP43, where it makes critical contacts with substrate (*Jaru-Ampornpan et al., 2013*; *Jaru-Ampornpan et al., 2010*; *McAvoy et al., 2018*; *Nguyen et al.,*

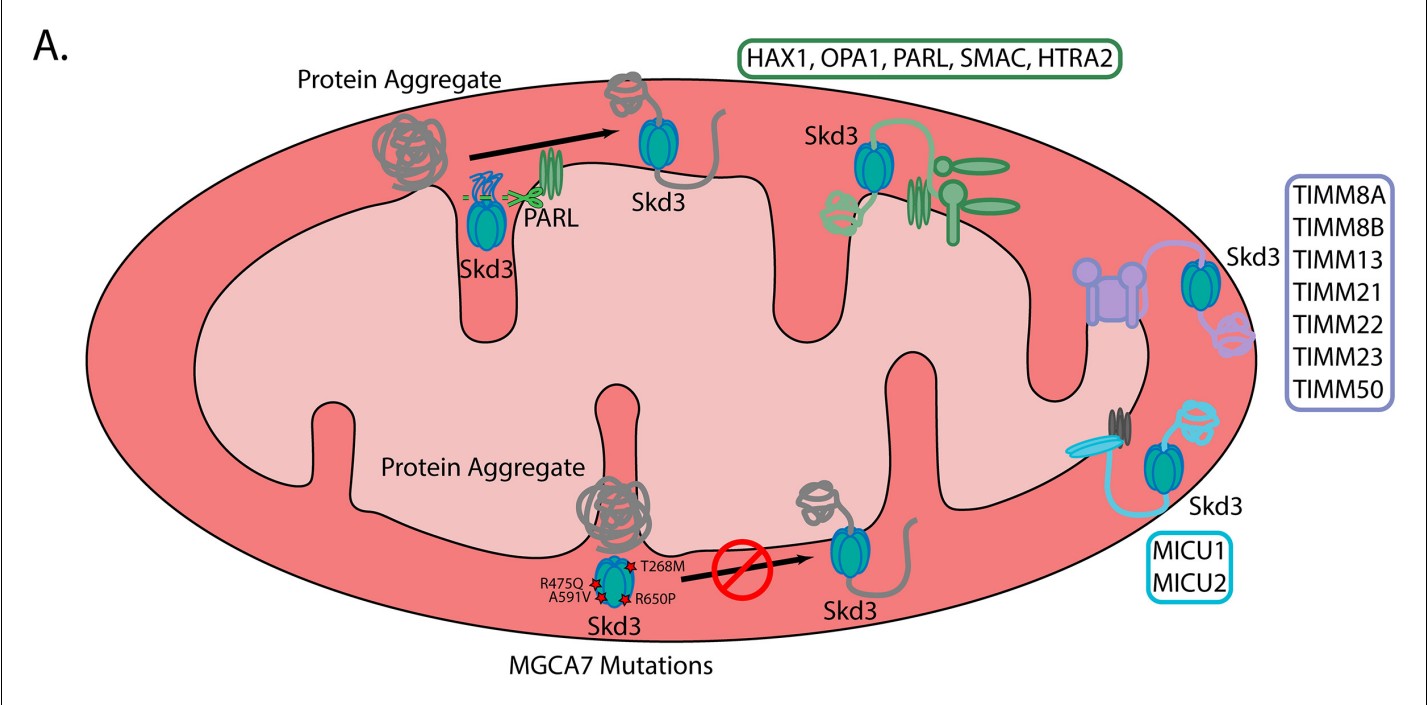

**Figure 9.** Skd3 is a protein disaggregase that is activated by PARL and inactivated by MGCA7-linked mutations. (**A**) Schematic illustrating (**i**) that Skd3 is a protein disaggregase that is activated by PARL cleavage of a hydrophobic auto-inhibitory peptide, (**ii**) that Skd3 works to solubilize key substrates in the mitochondrial intermembrane space and inner membrane that are involved in apoptosis, protein import, calcium handling, and respiration, and (**iii**) that mutations in Skd3 associated with MGCA7 result in defective Skd3 disaggregase activity in a manner that predicts the clinical severity of disease.

*2013*). Thus, ankyrin-repeat domains appear to be a recurring feature of diverse protein disaggregases.

Importantly, Skd3 is a stand-alone disaggregase and does not require Hsp70 and Hsp40 for maximal activity. This finding is consistent with the absence of Hsp70-interacting domains (NTD, NBD1, and MD) found in Hsp104, which enable collaboration with Hsp70 (*DeSantis et al., 2014*; *Lee et al., 2013*; *Sweeny et al., 2015*; *Sweeny et al., 2020*). Future structural and biochemical studies will further inform our mechanistic understanding of Skd3 disaggregase activity.

We establish that Skd3 can disaggregate disease-causing α-synuclein fibrils in vitro, demonstrating its robust activity as a disaggregase and identifying it as a potential therapeutic for synucleinopathies. The realization that human cells harbor a AAA+ protein disaggregase of greater potency than Hsp104 opens several therapeutic opportunities. Indeed, Skd3 is expressed in neurons and shifting localization of activated Skd3 to the cytoplasm could help combat cytoplasmic aggregates. Likewise, the expression of the PARLSkd3 enhanced variant in the cytoplasm of dopaminergic neurons may elicit therapeutic benefit similar to Hsp104 and engineered variants in Parkinson's disease models (*Jackrel et al., 2014*; *Lo Bianco et al., 2008*; *Mack et al., 2020*; *March et al., 2020*; *Tariq et al., 2019*). Future studies will further inform our understanding of how to harness Skd3 disaggregase activity therapeutically in synucleinopathies such as Parkinson's disease and other neurodegenerative diseases connected with aberrant protein aggregation.

We demonstrate that Skd3 is essential for maintaining the solubility of mitochondrial inner-membrane and intermembrane space protein complexes and specifically maintains solubility of the anti-apoptotic protein HAX1 in human cells (*Figure 9*). We suggest that HAX1 solubility is important for its anti-apoptotic effect. The precise mechanism of regulation between Skd3 and HAX1, PARL, OPA1, HTRA2, and SMAC/DIABLO warrants future study (*Figure 9*).

In addition to finding that many human mitochondrial proteins are more insoluble in the absence of Skd3, a small fraction of proteins are more insoluble in the presence of Skd3. Closer analysis of these enriched proteins suggests that many are mitochondrial matrix-associated, especially mitoribosome proteins (*Figure 7b*, *Supplementary file 1*). The mitoribosome is a large, megadalton sized

protein complex that is much more proteinaceous than its cytoplasmic counterpart and assembles into larger polysomes during active translation (*Couvillion et al., 2016*; *Greber and Ban, 2016*; *Saurer et al., 2019*). Thus, changes in solubility of the mitoribosome components could be due to increased mitoribosome or polysome assembly in the presence of Skd3.

It is surprising that Skd3 maintains protein solubility in the mitochondrial intermembrane space and inner membrane, as Hsp78 is found in the mitochondrial matrix (*Bateman et al., 2002*; *Germaniuk et al., 2002*; *Moczko et al., 1995*; *Rottgers et al., 2002*; *Schmitt et al., 1995*; *von Janowsky et al., 2006*). Since Skd3 appears in evolution alongside Hsp78 in choanoflagellates it may have initially arisen to serve a distinct function. We hypothesize that the increasing number and complexity of mitochondrial inner membrane protein assemblies (such as MICU1/MICU2/MCU and respiratory complex I) in metazoa might necessitate the requirement of Skd3 activity in the inner mitochondrial membrane and intermembrane space to maintain proteostasis in these compartments (*Botham et al., 2019*; *Hung et al., 2014*; *Yoshinaka et al., 2019*).

Hsp78 is known to be involved in mitochondrial protein import in yeast (*Moczko et al., 1995*; *Schmitt et al., 1995*). Furthermore, the ankyrin-repeat containing chloroplast disaggregase, cpSRP43, is involved in sorting membrane-targeted cargo (*Jaru-Ampornpan et al., 2013*; *Jaru-Ampornpan et al., 2010*). It is unknown if Skd3 participates in protein import and sorting in mitochondria. However, given the involvement of two closely related organelle-specific disaggregases in protein import and sorting and the changes in solubility of key TIM components in the absence of Skd3, it is plausible that Skd3 might play a role in protein import and sorting in mitochondria (*Figure 7b,c*). Mitochondrial presequences are prone to aggregation and the accumulation of uncleaved precursor aggregates can lead to the activation of an early mitochondrial unfolded protein response (mtUPR) (*Endo et al., 1995*; *Poveda-Huertes et al., 2020*; *Yano et al., 2003*). Skd3 might be required to keep aggregation-prone presequence signals in an unfolded state during import and sorting prior to cleavage or may help disaggregate them if they aggregate. Further investigation is required to determine which changes in protein solubility are driven by defects in Skd3 activity and which are from potential defects in mitochondrial protein import or membrane insertion.

Mutations in Skd3 are connected to MGCA7, which can be a devastating disorder involving severe neurologic deterioration, neutropenia, and death in infants (*Wortmann et al., 2016*). Importantly, we establish that diverse MGCA7-linked mutations in Skd3 impair disaggregase activity, but not necessarily ATPase activity (*Figure 9*). The degree of impaired disaggregase activity predicts the clinical severity of the disease, which suggests that disaggregase activity is a critical factor in disease. However, it is yet unclear which Skd3 substrate or substrates contribute to the MGCA7 etiology. Our mass-spectrometry data suggest that MGCA7 arises due to severely compromised proteostasis in the mitochondrial inner-membrane and intermembrane space (*Figure 9*). Hence, small-molecule drugs that restore wild-type levels of disaggregase activity to MGCA7-linked Skd3 variants could be valuable therapeutics.

Finally, Skd3 has also emerged as a factor in Venetoclax resistance, a FDA-approved drug for the treatment of acute myeloid leukemia (AML), which exerts its mechanism via BCL-2 inhibition (*Chen et al., 2019*). These studies suggest that inhibition of Skd3 may be of critical therapeutic importance for treating Venetoclax-resistant cancers (*Chen et al., 2019*). Small-molecule screens targeted at finding inhibitors of Skd3 disaggregase activity may yield important drugs for Venetoclax-resistant AML patients. Thus, the Skd3 disaggregase assays established in this study could provide a powerful platform for isolating therapeutic compounds for MGCA7 and AML.

## Materials and methods

### Multiple sequence alignments

NBD sequences were acquired via UniProtKB for *Homo sapiens* Skd3, *Saccharomyces cerevisiae* Hsp104, *Saccharomyces cerevisiae* Hsp78, *Escherichia coli* ClpB, *Escherichia coli* ClpA, and *Staphylococcus aureus* ClpC. Skd3 sequences were acquired via SMART protein domain annotation resource (*Letunic and Bork, 2018*). Sequences from *Anolis carolinensis*, *Bos taurus*, *Callithrix jacchus*, *Canis lupus*, *Capra hircus*, *Danio rerio*, *Equus caballus*, *Geospiza fortis*, *Gorilla gorilla*, *Homo sapiens*, *Monosiga brevicollis*, *Mus musculus*, *Nothobranchius rachovii*, *Rattus norvegicus*, *Sus scrofa*,

*Trachymyrmex septentrionalis*, *Trichinella papuae*, and *Xenopus laevis* were used to generate alignment for *Figure 1* and *Supplementary file 1*. Compiled sequences were aligned and made into a guide tree via Clustal Omega (*Madeira et al., 2019*). Alignment image was generated via BoxShade tool (*Hofmann and Baron, 1996*). Guide tree image was built using FigTree (*Rambaut, 2012*). Species images were used under license via PhyloPic. Sequence logo was created using WebLogo and 42 mammalian Skd3 protein sequences (*Ailuropoda melanoleuca*, *Callorhinus ursinus*, *Canis lupus*, *Carlito syrichta*, *Cebus capucinus*, *Ceratotherium simum*, *Cercocebus atys*, *Chlorocebus sabaeus*, *Colobus angolensis*, *Equus asinus*, *Equus caballus*, *Equus przewalskii*, *Felis catus*, *Gorilla gorilla*, *Gulo gulo*, *Grammomys surdaster*, *Homo sapiens*, *Macaca fascicularis*, *Macaca mulatta*, *Macaca nemestrina*, *Mandrillus leucophaeus*, *Microcebus murinus*, *Microtus ochrogaster*, *Mustela putorius*, *Nomascus leucogenys*, *Odobenus rosmarus*, *Orycteropus afer*, *Pan paniscus*, *Pan troglodyte*, *Panthera tigris*, *Papio anubis*, *Piliocolobus tephrosceles*, *Pongo abelii*, *Propithecus coquereli*, *Puma concolor*, *Rhinopithecus bieti*, *Rhinopithecus roxellana*, *Rousettus aegyptiacus*, *Theropithecus gelada*, *Ursus arctos*, *Ursus maritimus*, and *Zalophus californianus*) (*Crooks et al., 2004*; *Schneider and Stephens, 1990*).

## Cloning MBP-Skd3 plasmids

$_{MPP}$Skd3, $_{PARL}$Skd3, $_{ANK}$Skd3, and $_{NBD2}$Skd3 were cloned into the pMAL C2 plasmid with TEV site (*Yoshizawa et al., 2018*) using Gibson assembly (*Gibson et al., 2009*). The mitochondrial targeting signal was identified using MitoProt in agreement with previous work (*Claros and Vincens, 1996*; *Wortmann et al., 2015*). Site-directed mutagenesis was performed using QuikChange site-directed mutagenesis (Agilent) and confirmed by DNA sequencing.

## Purification of Skd3

Skd3 variants were expressed as an N-terminally MBP-tagged protein in BL21 (DE3) RIL cells (Agilent). Cells were lysed via sonication in 40 mM HEPES-KOH pH = 7.4, 500 mM KCl, 20% (w/v) glycerol, 5 mM ATP, 10 mM $MgCl_2$, 2 mM β-mercaptoethanol, 2.5 µM PepstatinA, and cOmplete Protease Inhibitor Cocktail (one tablet/250 mL, Millipore Sigma). Lysates were centrifuged at 30,597xg and 4℃ for 20 min and the supernatant was applied to amylose resin (NEB). The column was washed with 15 column volumes (CV) of wash buffer (WB: 40 mM HEPES-KOH pH = 7.4, 500 mM KCl, 20% (w/v) glycerol, 5 mM ATP, 10 mM $MgCl_2$, 2 mM β-mercaptoethanol, 2.5 µM PepstatinA, and cOmplete Protease Inhibitor Cocktail) at 4℃, 3 CV of WB supplemented with 20 mM ATP at 25℃ for 30 min, and 15 CV of WB at 4℃. The protein was then exchanged into elution buffer (EB: 50 mM Tris-HCl pH = 8.0, 300 mM KCl, 10% glycerol, 5 mM ATP, 10 mM $MgCl_2$, and 2 mM β-mercaptoethanol) with 8 CV and eluted via TEV cleavage at 34℃. The protein was then run over a size exclusion column (GE Healthcare HiPrep 26/60 Sephacryl S-300 HR) in sizing buffer (50 mM Tris-HCl pH = 8.0, 500 mM KCl, 10% glycerol, 1 mM ATP, 10 mM $MgCl_2$, and 1 mM DTT). Peak fractions were collected, concentrated to >5 mg/mL, supplemented with 5 mM ATP, and snap frozen. Protein purity was determined to be >95% by SDS-PAGE and Coomassie staining.

## Purification of Hsp104

Hsp104 was purified as previously described (*DeSantis et al., 2014*). In brief, Hsp104 was expressed in BL21 (DE3) RIL cells, lysed via sonication in lysis buffer [50 mM Tris-HCl pH = 8.0, 10 mM $MgCl_2$, 2.5% glycerol, 2 mM β-mercaptoethanol, 2.5 µM PepstatinA, and cOmplete Protease Inhibitor Cocktail (one mini EDTA-free tablet/50 mL, Millipore Sigma)], clarified via centrifugation at 30,597xg and 4℃ for 20 min, and purified on Affi-Gel Blue Gel (Bio-Rad). Hsp104 was eluted in elution buffer (50 mM Tris-HCl pH = 8.0, 1M KCl, 10 mM $MgCl_2$, 2.5% glycerol, and 2 mM β-mercaptoethanol) and then exchanged into storage buffer (40 mM HEPES-KOH pH = 7.4, 500 mM KCl, 20 mM $MgCl_2$, 10% glycerol, 1 mM DTT). The protein was diluted to 10% in buffer Q (20 mM Tris-HCl pH = 8.0, 50 mM NaCl, 5 mM $MgCl_2$, and 0.5 mM EDTA) and loaded onto a 5 mL RESOURCE Q anion exchange chromatography (GE Healthcare). Hsp104 was eluted via linear gradient of buffer Q+ (20 mM Tris pH = 8.0, 1M NaCl, 5 mM $MgCl_2$, and 0.5 mM EDTA). The protein was then exchanged into storage buffer and snap frozen. Protein purity was determined to be >95% by SDS-PAGE and Coomassie staining.

## Purification of Hsc70 and Hdj1

Hsc70 and Hdj1 were purified as previously described (*Michalska et al., 2019*). Hsc70 and Hdj1 were expressed in BL21 (DE3) RIL cells with an N-terminal His-SUMO tag. Cells were lysed via sonication into lysis buffer [50 mM HEPES-KOH pH = 7.5, 750 mM KCl, 5 mM $MgCl_2$, 10% glycerol, 20 mM imidazole, 2 mM β-mercaptoethanol, 5 µM pepstatin A, and cOmplete Protease Inhibitor Cocktail (one mini EDTA-free tablet/50 mL)]. Lysates were centrifuged at 30,597xg and 4°C for 20 min and the supernatant was bound to Ni-NTA Agarose resin (Qiagen), washed with 10 CV of wash buffer (50 mM HEPES-KOH pH = 7.5, 750 mM KCl, 10 mM $MgCl_2$, 10% glycerol, 20 mM imidazole, 1 mM ATP, and 2 mM β-mercaptoethanol), and then eluted with 2 CV of elution buffer (wash buffer supplemented with 300 mM imidazole). The tag was removed via Ulp1 (1:100 Ulp1:Protein molar ratio) cleavage during dialysis into wash buffer. The protein was further purified via loading onto a 5 mL HisTrap HP column (GE Healthcare) and pooling the untagged elution. Cleaved protein was pooled, concentrated, purified further by Resource Q ion exchange chromatography, and snap frozen. Protein purity was determined to be >95% by SDS-PAGE and Coomassie staining.

## ATPase assays

Proteins (0.25 µM monomer) were incubated with ATP (1 mM) (Innova Biosciences) at 37°C for 5 min (or otherwise indicated) in luciferase reactivation buffer (LRB; 25 mM HEPES-KOH [pH = 8.0], 150 mM KAOc, 10 mM MgAOc, 10 mM DTT). For substrate-stimulation of ATPase activity the indicated concentration of substrate was added. ATPase activity was assessed via inorganic phosphate release with a malachite green detection assay (Expedeon) and measured in Nunc 96 Well Optical plates on a Tecan Infinite M1000 plate reader. Background hydrolysis was measured at time zero and subtracted (*DeSantis et al., 2012*).

## Luciferase disaggregation and reactivation assays

Firefly luciferase aggregates were formed by incubating luciferase (50 µM) in LRB (pH=7.4) plus 8M urea at 30°C for 30 min. The luciferase was then rapidly diluted 100-fold into LRB, snap frozen, and stored at −80°C until use. Hsp104 and Skd3 variants (1 µM monomer, unless otherwise indicated) were incubated with 50 nM aggregated firefly luciferase in the presence or absence of Hsc70 and Hdj2 (0.167 µM each) in LRB plus 5 mM ATP plus an ATP regeneration system (ARS; 1 mM creatine phosphate and 0.25 µM creatine kinase) at 37°C for 90 min (unless otherwise indicated). The nucleotide-dependence of Skd3 disaggregation activity was tested in the presence of ATP (Sigma), AMP-PNP (Roche), ATPγS (Roche), or ADP (MP Biomedicals) for 30 min at 37°C without ARS. Recovered luminescence was monitored in Nunc 96 Well Optical plates using a Tecan Infinite M1000 plate reader (*DeSantis et al., 2012*).

## α-Synuclein disaggregation assay

α-Synuclein fibrils were acquired from the Luk lab and formed as previously described (*Luk et al., 2012*). $_{PARL}$Skd3 (10 µM monomer) was incubated with α-synuclein fibrils (0.5 µM monomer) in LRB in the presence or absence of ARS (10 mM ATP, 10 mM creatine phosphate, 40 µg/mL creatine kinase) at 37°C for 90 min. The samples were then centrifuged at 4°C and 20,000xg for 20 min. After centrifugation the supernatants were pipetted off of the pellets and the pellets were boiled in Pellet Buffer (PB; 50 mM Tris-HCl [pH = 8.0], 8M Urea, 150 mM NaCl, 10 uL/1 mL mammalian PI cocktail [Sigma CAT# P8340]) for 5 min at 99°C. The total sample, supernatant, and pellet samples were then blotted on nitrocellulose membrane (ThermoFisher Scientific CAT# 88018) and incubated with the SYN211 antibody (ThermoFisher Scientific CAT# AHB0261). Blots were then incubated with the IRDye 800CW Goat anti-Mouse IgG Secondary Antibody (Li-COR CAT# 926–32210) and imaged using the Li-Cor Odyssey Fc Imaging System. Samples were quantified using FIJI and normalized as (signal in supernatant)/(signal in pellet + signal in supernatant).

## Western blots

Mammalian whole cell lysates were prepared by boiling 500,000 cells in 1x Sample Buffer (SB; 60 mM Tris-HCl [pH = 6.8], 5% glycerol, 2% SDS, 10% β-mercaptoethanol, 0.025% bromophenol blue, 1x Mammalian PI cocktail) for 5 min at 99°C. Sedimentation assay samples were prepared as described above. Western blot samples were boiled for 5 min at 99°C in 1x SB, separated by SDS-

PAGE on a gradient gel (4%–20%, Bio-Rad CAT# 3450033), and transferred to a PVDF membrane. Membranes were blocked in Odyssey Blocking Buffer in PBS (Li-Cor CAT# 927–40000) for 1 hr at 25°C. Blots were then incubated in primary antibody overnight at 4°C and then in secondary for 30 min at 25°C. The antibodies used were: α-CLPB (Abcam CAT# ab235349), α-HAX1 (Abcam CAT# ab137613), α-COXIV (Abcam CAT# ab14744), α-GAPDH (Abcam CAT# ab8245), α-MICU2 (Abcam CAT# ab101465), IRDye 800CW Goat α-mouse secondary (Li-Cor CAT# 926–32210), and IRDye 680RD Goat α-rabbit secondary (Li-Cor CAT# 926–68071). Blots were imaged on a Li-Cor Odyssey Fc Imaging System.

## Mammalian cell culture

Isogenic HAP1 and HAP1 ΔCLPB cells were acquired directly from Horizon Discovery (CAT# HZGHC81570 and HZGHC007326c001) and have been control quality checked by the vendor. HAP1 cells are a human near-haploid cell line derived from the male chronic myelogenous leukemia (CML) cell line KBM-7 (*Essletzbichler et al., 2014*). In brief, the *CLPB* knockout line was generated via CRISPR-Cas9 and the gRNA TGGCACGGGCAGCTTCCAAC to make a 1 bp insertion into exon 2 of the *CLPB* gene. Knockout was confirmed by Sanger sequencing (DNA) by the vendor and western blot (*Figure 7—figure supplement 1*). Cells were maintained in IMDM (Gibco CAT# 12440053) supplemented with 10% FBS (GE CAT# SH3007003) and 1% P/S (Gibco CAT# 15140122) at 37°C and 5% $CO_2$. Cells were grown at a confluency of 50–60% for mitochondrial isolation.

## Mitochondrial isolation

Mitochondria were isolated as previously described (*Frezza et al., 2007*). In brief, $50–100*10^6$ cells were resuspended in 5 mL SM buffer (50 mM Tris-HCl [pH = 7.4], 0.25M sucrose, 2 mM EDTA, and 1% BSA) and homogenized with a Dounce homogenizer and Teflon pestle (30 strokes at 600 RPM) at 4°C. Lysate was then centrifuged at 600xg for 10 min. The supernatant was collected, and the pellet was dissolved in 5 mL SM buffer and homogenized (15 strokes at 600 RPM). Sample was then centrifuged at 600xg for 10 min and the supernatant was pooled and centrifuged at 12,000xg for 15 min. The pellet was collected and used for further experiments.

## Mitochondrial sedimentation assay

Mitochondrial sedimentation assay was performed essentially as previously described (*Wilkening et al., 2018*). 60–80 μg isolated mitochondria were resuspended in 200 μL Mitochondrial Resuspension Buffer (40 mM HEPES-KOH, pH = 7.6, 500 mM sucrose, 120 mM K-acetate, 10 mM Mg-acetate, 5 mM glutamate, 5 mM malate, 5 mM EDTA, 5 mM ATP, 20 mM creatine phosphate, 4 μg/mL creatine kinase, 1 mM DTT) and incubated at 37°C for 20 min. The mitochondria were then pelleted at 12,000xg for 10 min at 4°C. The mitochondria were then resuspended in 200 μL Lysis Buffer (30 mM Tris-HCl, pH = 7.4, 200 mM KCl, 0.5% v/v Triton X-100, 5 mM EDTA, 0.5 mM PMSF, 1x Mammalian PI cocktail) and lysed in a thermomixer at 2,000 RPM for 10 min at 4°C. The protein concentration of the lysate was then quantified using a BCA assay (ThermoFisher CAT# 23225). 12 μg of lysate was added to a total volume of 50 μL Lysis Buffer and reserved as a total protein sample. 12 μg of lysate was added to a total volume of 50 μL Lysis Buffer and sedimented at 20,000xg for 20 min at 4°C. The supernatant was removed, TCA precipitated, and frozen for later processing. The pellet was boiled in 10 μL of Pellet Buffer and frozen for later processing.

## Mass spectrometry

Pellet samples were excised as whole lanes from gels, reduced with TCEP, alkylated with iodoacetamide, and digested with trypsin. Tryptic digests were desalted by loading onto a MonoCap C18 Trap Column (GL Sciences), flushed for 5 min at 6 μL/min using 100% Buffer A ($H_2O$, 0.1% formic acid), then analyzed via LC (Waters NanoAcquity) gradient using Buffer A and Buffer B (acetonitrile, 0.1% formic acid) (initial 5% B; 75 min 30% B; 80 min 80% B; 90.5–105 min 5% B) on the Thermo Q Exactive HF mass spectrometer. Data were acquired in data-dependent mode. Analysis was performed with the following settings: MS1 60K resolution, AGC target 3e6, max inject time 50 ms; MS2 Top N = 20 15K resolution, AGC target 5e4, max inject time 50 ms, isolation window = 1.5 m/z, normalized collision energy 28%. LC-MS/MS data were searched with full tryptic specificity against the UniProt human database using MaxQuant 1.6.8.0. MS data were also searched for common

protein N-terminal acetylation and methionine oxidation. Protein and peptide false discovery rate was set at 1%. LFQ intensity was calculated using the MaxLFQ algorithm (*Cox et al., 2014*). Fold enrichment was calculated as LFQ intensity from the ΔCLPB pellet divided by the LFQ intensity from the wild-type pellet. High confidence hits were quantified as minimum absolute fold change of 2 and p-value<0.05.

## Acknowledgements

We thank Kelvin Luk, Douglas C Wallace, Prasanth Potluri, and Manisha Koneru for generously providing key reagents. We thank the Wistar Institute Proteomics and Metabolomics Facility for their assistance with the LC-MS/MS experiments, especially Hsin-Yao Tang and Thomas Beer. We thank Edward Barbieri, Edward Chuang, Kelvin Luk, Jordan Cupo, Albert Erives, Jan Fassler, Charlotte Fare, Oliver King, JiaBei Lin, Dylan Marchione, Zachary March, Hana Odeh, April Darling, and Bede Portz for feedback on the manuscript. Our work was funded by the Blavatnik Family Foundation Fellowship (RRC), The G Harold and Leila Y Mathers Foundation (JS), and NIH grants T32GM008275 (RRC), F31AG060672 (RRC), R01GM099836 (JS), and R21AG061784 (JS).

## Additional information

### Funding

| Funder | Grant reference number | Author |
| --- | --- | --- |
| National Institutes of Health | R01GM099836 | James Shorter |
| National Institutes of Health | F31AG060672 | Ryan R Cupo |
| National Institutes of Health | T32GM008275 | Ryan R Cupo |
| Blavatnik Family Foundation | Blavatnik Family Foundation Fellowship | Ryan R Cupo |
| Mathers Foundation | MF-1902-00275 | James Shorter |
| National Institutes of Health | R21AG061784 | James Shorter |

The funders had no role in study design, data collection and interpretation, or the decision to submit the work for publication.

### Author contributions

Ryan R Cupo, Conceptualization, Data curation, Funding acquisition, Validation, Investigation, Visualization, Methodology, Writing - original draft; James Shorter, Conceptualization, Resources, Supervision, Funding acquisition, Writing - original draft

### Author ORCIDs

Ryan R Cupo 🆔 https://orcid.org/0000-0001-7639-1923
James Shorter 🆔 https://orcid.org/0000-0001-5269-8533

### Decision letter and Author response

Decision letter https://doi.org/10.7554/eLife.55279.sa1
Author response https://doi.org/10.7554/eLife.55279.sa2

## Additional files

### Supplementary files

• Supplementary file 1. Alignment of Skd3 to diverse metazoan lineages shows conservation of key motifs and domains. Alignment of Skd3 protein from diverse metazoan lineages. Alignment was constructed using Clustal Omega. Alignment shows high level of conservation of Skd3 among species. *H. sapiens*, *G. gorilla*, and *C. jacchus* Skd3 have an additional insertion in the ankyrin-repeat domain that is not conserved in the other species. This alignment was used to generate the phylogenetic

tree in *Figure 1B*. The protozoan *M. brevicollis* Skd3 sequence was included in the alignment for reference. MTS (mitochondrial-targeting sequence, ANK (ankyrin-repeat domain), NBD (nucleotide-binding domain), and CTD (C-terminal domain).

• Supplementary file 2. Proteins enriched in HAP1 Δ*CLPB* pellet. Proteins from mass spectrometry data in *Figure 7b* highlighted in red that have >2.0 fold change increase in the Δ*CLPB* insoluble fraction compared to wild type and a p-value of <0.05.

• Supplementary file 3. Proteins enriched in HAP1 WT pellet. Proteins from mass spectrometry data in *Figure 7b* highlighted in green that have >2.0 fold change increase in the wild-type insoluble fraction compared to Δ*CLPB* and a p-value of <0.05.

• Transparent reporting form

## Data availability

All data generated or analysed during this study are included in the manuscript and supporting files.

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

# Appendix 1

**Appendix 1—key resources table**

| Reagent type (species) or resource | Designation | Source or reference | Identifiers | Additional information |
|---|---|---|---|---|
| Antibody | Mouse monoclonal $\alpha$-$\alpha$-synuclein (SYN211) | ThermoFisher Scientific | AHB0261; RRID: AB_10978319 | (1:500 dilution) |
| Antibody | Rabbit polyclonal $\alpha$-CLPB (Skd3) | abcam | ab235349; RRID: AB_2847899 | (1:1000 dilution) |
| Antibody | Rabbit polyclonal $\alpha$-CLPB (Skd3) | abcam | ab76179; RRID:AB_1310087 | (1:1000 dilution) |
| Antibody | Rabbit polyclonal $\alpha$-CLPB (Skd3) | abcam | ab87253; RRID:AB_1952530 | (1:1000 dilution) |
| Antibody | Rabbit polyclonal $\alpha$-CLPB (Skd3) | Proteintech | 15743–1-AP; RRID: AB_2847900 | (1:1000 dilution) |
| Antibody | Mouse monoclonal $\alpha$-COXIV | abcam | ab14744; RRID:AB_301443 | (1:1000 dilution) |
| Antibody | Mouse monoclonal $\alpha$-GAPDH | abcam | ab8245; RRID:AB_2107448 | (1:1000 dilution) |
| Antibody | Rabbit polyclonal $\alpha$-HAX1 | abcam | ab137613; RRID: AB_2847902 | (1:500 dilution) |
| Antibody | Rabbit polyclonal $\alpha$-MICU2 | abcam | ab101465; RRID: AB_10711219 | (1:1000 dilution) |
| Antibody | IRDye 680RD Goat anti-Rabbit IgG Secondary Antibody | Li-Cor | 926–68071; RRID: AB_10956166 | (1:2500 dilution) |
| Antibody | IRDye 800CW Goat anti-Mouse IgG Secondary Antibody | Li-Cor | 926–32210; RRID: AB_621842 | (1:2500 dilution) |
| Cell line (*Escherichia coli*) | BL21-CodonPlus (DE3)-RIL competent cells | Agilent | 230245 | Chemically competent cells |
| Cell line (*Homo sapiens*) | Human HAP1 Knockout Cell Lines - WT | Horizon Discovery | HZGHC81570;; RRID:CVCL_Y019 | Human control cell line. The HAP1 cell line was bought directly from Horizon Discovery and has been control quality checked by the vendor. |
| Cell line (*Homo sapiens*) | Human HAP1 Knockout Cell Lines - CLPB | Horizon Discovery | HZGHC007326c001 | Human knockout cell line. The HAP1 ClpB knockout cell line was bought directly from Horizon Discovery and has been control quality checked by the vendor. |
| Commercial assay or kit | ATPase Activity Kit (Colorimetric) | Innova Biosciencens | 601–0120 | |
| Commercial assay or kit | Luciferase Assay Reagent | Promega | E1483 | |
| Chemical compound, drug | Creatine phosphate | Roche | 10621722001 | |

*Appendix 1—key resources table continued*

| Reagent type (species) or resource | Designation | Source or reference | Identifiers | Additional information |
|---|---|---|---|---|
| Chemical compound, drug | ADP | MP Biomedicals | 150260 | |
| Chemical compound, drug | AMPPNP | Roche | 10102547001 | |
| Chemical compound, drug | ATP | Sigma-Aldrich | A3377 | |
| Chemical compound, drug | ATP$\gamma$S | Roche | 10102342001 | |
| Chemical compound, drug | cOmplete, Mini, EDTA-free Protease Inhibitor Cocktail | Sigma-Aldrich | 4693159001 | |
| Chemical compound, drug | Pepstatin A, microbial,$\geq$90% (HPLC) | Sigma-Aldrich | P5318 | |
| Chemical compound, drug | Protease Inhibitor Cocktail (mammalian) | Sigma-Aldrich | P8340 | |
| Gene (*Homo sapiens*) | *CLPB* | NA | HGNC:30664 | |
| Peptide, recombinant protein | $\alpha$-synuclein fibrils | *Luk et al., 2012* | | Preformed $\alpha$-synuclein fibrils were a gift from Kelvin Luk. |
| Peptide, recombinant protein | $\beta$-casein | Sigma-Aldrich | C6905 | |
| Peptide, recombinant protein | Creatine kinase | Roche | 10127566001 | |
| Peptide, recombinant protein | Firefly luciferase | Sigma-Aldrich | L9506 | |
| Peptide, recombinant protein | Hdj1 | *Michalska et al., 2019* | | |
| Peptide, recombinant protein | Hsc70 | *Michalska et al., 2019* | | |
| Peptide, recombinant protein | Hsp104 | *DeSantis et al., 2012* and *Jackrel et al., 2014* | | |
| Peptide, recombinant protein | Lysozyme | Sigma-Aldrich | L6876 | |
| Peptide, recombinant protein | MPPSkd3 | This study | | Full description can be found in Materials and methods: Purification of Skd3 |

*Appendix 1—key resources table continued*

| Reagent type (species) or resource | Designation | Source or reference | Identifiers | Additional information |
|---|---|---|---|---|
| Peptide, re-combinant protein | $_{MPP}$Skd3$^{K387A}$ | This study | | Full description can be found in Materials and methods: Purification of Skd3 |
| Peptide, re-combinant protein | $_{MPP}$Skd3$^{E455Q}$ | This study | | Full description can be found in Materials and methods: Purification of Skd3 |
| Peptide, re-combinant protein | $_{MPP}$Skd3$^{Y430A}$ | This study | | Full description can be found in Materials and methods: Purification of Skd3 |
| Peptide, re-combinant protein | $_{PARL}$Skd3 | This study | | Full description can be found in Materials and methods: Purification of Skd3 |
| Peptide, re-combinant protein | $_{PARL}$Skd3$^{K387A}$ | This study | | Full description can be found in Materials and methods: Purification of Skd3 |
| Peptide, re-combinant protein | $_{PARL}$Skd3$^{E455Q}$ | This study | | Full description can be found in Materials and methods: Purification of Skd3 |
| Peptide, re-combinant protein | $_{PARL}$Skd3$^{Y430A}$ | This study | | Full description can be found in Materials and methods: Purification of Skd3 |
| Peptide, re-combinant protein | $_{PARL}$Skd3$^{T268M}$ | This study | | Full description can be found in Materials and methods: Purification of Skd3 |
| Peptide, re-combinant protein | $_{PARL}$Skd3$^{R475Q}$ | This study | | Full description can be found in Materials and methods: Purification of Skd3 |
| Peptide, re-combinant protein | $_{PARL}$Skd3$^{A591V}$ | This study | | Full description can be found in Materials and methods: Purification of Skd3 |
| Peptide, re-combinant protein | $_{PARL}$Skd3$^{R650P}$ | This study | | Full description can be found in Materials and methods: Purification of Skd3 |
| Peptide, re-combinant protein | Skd3$_{ANK}$ | This study | | Full description can be found in Materials and methods: Purification of Skd3 |
| Peptide, re-combinant protein | Skd3$_{NBD}$ | This study | | Full description can be found in Materials and methods: Purification of Skd3 |
| Recombinant DNA reagent | Hdj1 in pE-SUMO | *Michalska et al., 2019* | | |
| Recombinant DNA reagent | Hsc70 in pE-SUMO | *Michalska et al., 2019* | | |

*Appendix 1—key resources table continued*

| Reagent type (species) or resource | Designation | Source or reference | Identifiers | Additional information |
|---|---|---|---|---|
| Recombinant DNA reagent | Hsp104 in pNO-TAG | *DeSantis et al., 2012* and *Jackrel et al., 2014* | | |
| Recombinant DNA reagent | $_{MPP}$Skd3 in pMAL C2 with TEV site | This study | | Full description can be found in Materials and methods: Purification of Skd3 |
| Recombinant DNA reagent | $_{MPP}$Skd3$^{K387A}$ in pMAL C2 with TEV site | This study | | Full description can be found in Materials and methods: Purification of Skd3 |
| Recombinant DNA reagent | $_{MPP}$Skd3$^{E455Q}$ in pMAL C2 with TEV site | This study | | Full description can be found in Materials and methods: Purification of Skd3 |
| Recombinant DNA reagent | $_{MPP}$Skd3$^{Y430A}$ in pMAL C2 with TEV site | This study | | Full description can be found in Materials and methods: Purification of Skd3 |
| Recombinant DNA reagent | $_{PARL}$Skd3 in pMAL C2 with TEV site | This study | | Full description can be found in Materials and methods: Purification of Skd3 |
| Recombinant DNA reagent | $_{PARL}$Skd3$^{K387A}$ in pMAL C2 with TEV site | This study | | Full description can be found in Materials and methods: Purification of Skd3 |
| Recombinant DNA reagent | $_{PARL}$Skd3$^{E455Q}$ in pMAL C2 with TEV site | This study | | Full description can be found in Materials and methods: Purification of Skd3 |
| Recombinant DNA reagent | $_{PARL}$Skd3$^{Y430A}$ in pMAL C2 with TEV site | This study | | Full description can be found in Materials and methods: Purification of Skd3 |
| Recombinant DNA reagent | $_{PARL}$Skd3$^{T268M}$ in pMAL C2 with TEV site | This study | | Full description can be found in Materials and methods: Purification of Skd3 |
| Recombinant DNA reagent | $_{PARL}$Skd3$^{R475Q}$ in pMAL C2 with TEV site | This study | | Full description can be found in Materials and methods: Purification of Skd3 |
| Recombinant DNA reagent | $_{PARL}$Skd3$^{A591V}$ in pMAL C2 with TEV site | This study | | Full description can be found in Materials and methods: Purification of Skd3 |
| Recombinant DNA reagent | $_{PARL}$Skd3$^{R650P}$ in pMAL C2 with TEV site | This study | | Full description can be found in Materials and methods: Purification of Skd3 |
| Recombinant DNA reagent | Skd3$_{ANK}$ in pMAL C2 with TEV site | This study | | Full description can be found in Materials and methods: Purification of Skd3 |

*Appendix 1—key resources table continued*

| Reagent type (species) or resource | Designation | Source or reference | Identifiers | Additional information |
|---|---|---|---|---|
| Recombinant DNA reagent | Skd3$_{NBD}$ in pMAL C2 with TEV site | This study | | Full description can be found in Materials and methods: Purification of Skd3 |
| Sequence-based reagent | ClpA (*Escherichia coli*) | This study | UniProtKB:P0ABH9 | Full description can be found in Materials and methods: Purification of Skd3 |
| Sequence-based reagent | ClpB (*Escherichia coli*) | This study | UniProtKB:P63284 | Full description can be found in Materials and methods: Purification of Skd3 |
| Sequence-based reagent | ClpC (*Staphylococcus aureus*) | This study | UniProtKB:Q2G0P5 | Full description can be found in Materials and methods: Purification of Skd3 |
| Sequence-based reagent | Hsp104 (*Saccharomyces cerevisiae*) | This study | UniProtKB:P31539 | Full description can be found in Materials and methods: Purification of Skd3 |
| Sequence-based reagent | Hsp78 (*Saccharomyces cerevisiae*) | This study | UniProtKB:P33416 | Full description can be found in Materials and methods: Purification of Skd3 |
| Sequence-based reagent | HAX1 (*Homo sapiens*) | This study | UniProtKB:O00165 | Full description can be found in Materials and methods: Purification of Skd3 |
| Sequence-based reagent | Skd3 (*Ailuropoda melanoleuca*) | This study | | Full description can be found in Materials and methods: Purification of Skd3 |
| Sequence-based reagent | Skd3 (*Anolis carolinensis*) | This study | | Full description can be found in Materials and methods: Purification of Skd3 |
| Sequence-based reagent | Skd3 (*Bos taurus*) | This study | | Full description can be found in Materials and methods: Purification of Skd3 |
| Sequence-based reagent | Skd3 (*Callithrix jacchus*) | This study | | Full description can be found in Materials and methods: Purification of Skd3 |
| Sequence-based reagent | Skd3 (*Callorhinus ursinus*) | This study | | Full description can be found in Materials and methods: Purification of Skd3 |
| Sequence-based reagent | Skd3 (*Canis lupus*) | This study | | Full description can be found in Materials and methods: Purification of Skd3 |
| Sequence-based reagent | Skd3 (*Capra hircus*) | This study | | Full description can be found in Materials and methods: Purification of Skd3 |

*Appendix 1—key resources table continued*

| Reagent type (species) or resource | Designation | Source or reference | Identifiers | Additional information |
|---|---|---|---|---|
| Sequence-based reagent | Skd3 (*Carlito syrichta*) | This study | | Full description can be found in Materials and methods: Purification of Skd3 |
| Sequence-based reagent | Skd3 (*Cebus capucinus*) | This study | | Full description can be found in Materials and methods: Purification of Skd3 |
| Sequence-based reagent | Skd3 (*Ceratotherium simum*) | This study | | Full description can be found in Materials and methods: Purification of Skd3 |
| Sequence-based reagent | Skd3 (*Cercocebus atys*) | This study | | Full description can be found in Materials and methods: Purification of Skd3 |
| Sequence-based reagent | Skd3 (*Chlorocebus sabaeus*) | This study | | Full description can be found in Materials and methods: Purification of Skd3 |
| Sequence-based reagent | Skd3 (*Colobus angolensis*) | This study | | Full description can be found in Materials and methods: Purification of Skd3 |
| Sequence-based reagent | Skd3 (*Danio rerio*) | This study | | Full description can be found in Materials and methods: Purification of Skd3 |
| Sequence-based reagent | Skd3 (*Equus asinus*) | This study | | Full description can be found in Materials and methods: Purification of Skd3 |
| Sequence-based reagent | Skd3 (*Equus caballus*) | This study | | Full description can be found in Materials and methods: Purification of Skd3 |
| Sequence-based reagent | Skd3 (*Equus przewalskii*) | This study | | Full description can be found in Materials and methods: Purification of Skd3 |
| Sequence-based reagent | Skd3 (*Felis catus*) | This study | | Full description can be found in Materials and methods: Purification of Skd3 |
| Sequence-based reagent | Skd3 (*Geospiza fortis*) | This study | | Full description can be found in Materials and methods: Purification of Skd3 |
| Sequence-based reagent | Skd3 (*Gorilla gorilla*) | This study | | Full description can be found in Materials and methods: Purification of Skd3 |
| Sequence-based reagent | Skd3 (*Gulo gulo*) | This study | | Full description can be found in Materials and methods: Purification of Skd3 |

*Appendix 1—key resources table continued*

| Reagent type (species) or resource | Designation | Source or reference | Identifiers | Additional information |
|---|---|---|---|---|
| Sequence-based reagent | Skd3 (*Grammomys surdaster*) | This study | | Full description can be found in Materials and methods: Purification of Skd3 |
| Sequence-based reagent | Skd3 (*Homo sapiens*) | This study | UniProtKB:Q9H078 | Full description can be found in Materials and methods: Purification of Skd3 |
| Sequence-based reagent | Skd3 (*Macaca fascicularis*) | This study | | Full description can be found in Materials and methods: Purification of Skd3 |
| Sequence-based reagent | Skd3 (*Macaca mulatta*) | This study | | Full description can be found in Materials and methods: Purification of Skd3 |
| Sequence-based reagent | Skd3 (*Macaca nemestrina*) | This study | | Full description can be found in Materials and methods: Purification of Skd3 |
| Sequence-based reagent | Skd3 (*Mandrillus leucophaeus*) | This study | | Full description can be found in Materials and methods: Purification of Skd3 |
| Sequence-based reagent | Skd3 (*Microcebus murinus*) | This study | | Full description can be found in Materials and methods: Purification of Skd3 |
| Sequence-based reagent | Skd3 (*Microtus ochrogaster*) | This study | | Full description can be found in Materials and methods: Purification of Skd3 |
| Sequence-based reagent | Skd3 (*Monosigia brevicollis*) | This study | | Full description can be found in Materials and methods: Purification of Skd3 |
| Sequence-based reagent | Skd3 (*Mus musculus*) | This study | | Full description can be found in Materials and methods: Purification of Skd3 |
| Sequence-based reagent | Skd3 (*Mustela putorius*) | This study | | Full description can be found in Materials and methods: Purification of Skd3 |
| Sequence-based reagent | Skd3 (*Nomascus leucogenys*) | This study | | Full description can be found in Materials and methods: Purification of Skd3 |
| Sequence-based reagent | Skd3 (*Nothobranchius rachovii*) | This study | | Full description can be found in Materials and methods: Purification of Skd3 |
| Sequence-based reagent | Skd3 (*Odobenus rosmarus*) | This study | | Full description can be found in Materials and methods: Purification of Skd3 |

*Appendix 1—key resources table continued*

| Reagent type (species) or resource | Designation | Source or reference | Identifiers | Additional information |
|---|---|---|---|---|
| Sequence-based reagent | Skd3 (*Orycteropus afer*) | This study | | Full description can be found in Materials and methods: Purification of Skd3 |
| Sequence-based reagent | Skd3 (*Pan paniscus*) | This study | | Full description can be found in Materials and methods: Purification of Skd3 |
| Sequence-based reagent | Skd3 (*Pan troglodyte*) | This study | | Full description can be found in Materials and methods: Purification of Skd3 |
| Sequence-based reagent | Skd3 (*Panthera tigris*) | This study | | Full description can be found in Materials and methods: Purification of Skd3 |
| Sequence-based reagent | Skd3 (*Papio anubis*) | This study | | Full description can be found in Materials and methods: Purification of Skd3 |
| Sequence-based reagent | Skd3 (*Piliocolobus tephrosceles*) | This study | | Full description can be found in Materials and methods: Purification of Skd3 |
| Sequence-based reagent | Skd3 (*Pongo abelii*) | This study | | Full description can be found in Materials and methods: Purification of Skd3 |
| Sequence-based reagent | Skd3 (*Propithecus coquereli*) | This study | | Full description can be found in Materials and methods: Purification of Skd3 |
| Sequence-based reagent | Skd3 (*Puma concolor*) | This study | | Full description can be found in Materials and methods: Purification of Skd3 |
| Sequence-based reagent | Skd3 (*Rattus norvegicus*) | This study | | Full description can be found in Materials and methods: Purification of Skd3 |
| Sequence-based reagent | Skd3 (*Rhinopithecus bieti*) | This study | | Full description can be found in Materials and methods: Purification of Skd3 |
| Sequence-based reagent | Skd3 (*Rhinopithecus roxellana*) | This study | | Full description can be found in Materials and methods: Purification of Skd3 |
| Sequence-based reagent | Skd3 (*Rousettus aegyptiacus*) | This study | | Full description can be found in Materials and methods: Purification of Skd3 |
| Sequence-based reagent | Skd3 (*Sus scrofa*) | This study | | Full description can be found in Materials and methods: Purification of Skd3 |

*Appendix 1—key resources table continued*

| Reagent type (species) or resource | Designation | Source or reference | Identifiers | Additional information |
|---|---|---|---|---|
| Sequence-based reagent | Skd3 (*Theropithecus gelada*) | This study | | Full description can be found in Materials and methods: Purification of Skd3 |
| Sequence-based reagent | Skd3 (*Trachymyrmex septentrionalis*) | This study | | Full description can be found in Materials and methods: Purification of Skd3 |
| Sequence-based reagent | Skd3 (*Trichinella papuae*) | This study | | Full description can be found in Materials and methods: Purification of Skd3 |
| Sequence-based reagent | Skd3 (*Ursus arctos*) | This study | | Full description can be found in Materials and methods: Purification of Skd3 |
| Sequence-based reagent | Skd3 (*Ursus maritimus*) | This study | | Full description can be found in Materials and methods: Purification of Skd3 |
| Sequence-based reagent | Skd3 (*Xenopus laevis*) | This study | | Full description can be found in Materials and methods: Purification of Skd3 |
| Sequence-based reagent | Skd3 (*Zalophus californianus*) | This study | | Full description can be found in Materials and methods: Purification of Skd3 |
| Software, algorithm | BOXSHADE | *Hofmann and Baron, 1996* | | |
| Software, algorithm | Clusal Omega | *Madeira et al., 2019* | | |
| Software, algorithm | EXPASY ProtParam | *Kyte and Doolittle, 1982*; *Wilkins et al., 1999* | | |
| Software, algorithm | FigTree | *Rambaut, 2012* | | |
| Software, algorithm | Gene Ontology (GO) | *Ashburner et al., 2000*; *Mi et al., 2019*; *The Gene Ontology Consortium, 2019* | | |
| Software, algorithm | IUPRED | *Mészáros et al., 2018*; *Mészáros et al., 2009* | | |
| Software, algorithm | PhyloPic | http://phylopic.org/ | | |
| Software, algorithm | Prism 8 | GraphPad | | |
| Software, algorithm | WebLogo | https://weblogo.berkeley.edu/logo.cgi | | |

