## [Decision Letter]

**Acceptance summary:**

Skd3 is a mitochondrial AAA+ enzyme whose function has been poorly-understood. The authors reveal that Skd3 is capable of both reactivating a model aggregated luciferase substrate and solubilizing α-synuclein in vitro. The in vitro activity of Skd3 is shown to be enhanced by the proteolytic cleavage of an autoinhibitory peptide, which is removed by the PARL protease in mitochondria. Useful mechanistic insights are also provided into some of the known disease-associated mutations in Skd3. Overall, the findings are of significant interest, as they address long-standing questions concerning the function of Skd3 and how metazoans handle mitochondrial aggregation in the absence of Hsp110 and Hsp78.

**Decision letter after peer review:**

Thank you for submitting your article "Skd3 is a potent mitochondrial protein disaggregase that is inactivated by 3-methylglutaconic aciduria-linked mutations" for consideration by *eLife*. Your article has been reviewed by two peer reviewers, and the evaluation has been overseen by a Reviewing Editor and Huda Zoghbi as the Senior Editor. The following individual involved in review of your submission has agreed to reveal their identity: Steven Glynn (Reviewer #1).

The reviewers have discussed the reviews with one another and the Reviewing Editor has drafted this decision to help you prepare a revised submission.

Summary:

Skd3 is a mitochondrial AAA+ enzyme whose function has been poorly-understood. The authors reveal that Skd3 is capable of reactivating a model luciferase aggregate and solubilizing α-synuclein in vitro. The in vitro activity of Skd3 is shown to be enhanced by the proteolytic cleavage of an autoinhibitory peptide, which is removed by the PARL protease in mitochondria. Loss of SKD3 in human mitochondria is found to decrease the solubility of several specific proteins in the intermembrane space, although it is unclear whether this is a direct effect. The experiments employed are straight-forward and a full analysis of SKD3 activity, including the identify of its physiological substrates, will require further studies.

The study is carefully done, and the manuscript is well-written. Although a recent study reported a similar finding (i.e., that Skd3 serves as disaggregase), the present study is substantially more thorough and validates at least some of the in vitro findings in the cellular context. The work also provides useful mechanistic insights into some of the known disease-associated mutations in Skd3. Overall, the findings are of significant interest in addressing long-standing questions concerning the function of Skd3 and how metazoans handle mitochondrial aggregation in the absence of Hsp110 and Hsp78. This said, the study falls short on demonstrating true disaggregation activity in vitro and in vivo, behaviors that ideally would need to be validated before publication is warranted.

While this review was being finalized, the complexities of COVID-19 emerged. Given these recent developments, the reviewers recognize that the additional experiments they initially requested may not be possible. We would therefore ask that the authors first read the 'original' review for a sense of the issues raised by the reviewers, and then the 'updated' review, which follows for a revised set of requests.

Essential revisions:

1) The authors demonstrate that Skd3 reactivates thermally inactivated luciferase but do not show directly that Skd3 actually dissociates luciferase aggregates (the solubilization of α-synuclein in Figure 4 is insufficient to demonstrate such an activity because the experiments never start with the isolated fibrils only and no equivalent experiments are shown with luciferase). An in vitro test for disaggregation activity (e.g., spinning down and quantifying aggregates as a function of Skd3 activity) is necessary to validate the primary point of the manuscript.

2) Similar to (1), the authors should show that Skd3 is able to disaggregate proteins in vivo (the increased solubility of HAX1 in Figure 7 is insufficient to demonstrate such an activity because it cannot distinguish between whether Skd3 resolves aggregation or prevents aggregation from happening in the first place). One approach that might be of use is that of Janowsky et al., who showed that mitochondrial Hsp78 was to induce thermal aggregation of an imported protein in purified mitochondria and then monitor disaggregation +/- Hsp78 (von Janowsky et al., 2006). Other work by Liberek provides some alternative approaches.

3) The changes in mitochondrial protein solubility observed in SKD3 knockout cells could be caused by loss of SKD3 disaggregase activity on specific proteins or secondary defects in protein import or membrane integration leading to accumulation of misfolded/uninserted proteins in the IMS. The authors should address this question in the Discussion.

4) The loss of ATPase and disaggregase activity seen for the isolated NBD domain (SKD3NBD) could be explained by disassembly of SKD3 hexamers in the absence of the ankyrin domains. The authors should include information on whether these constructs form stable hexamers.

5) Figures 2C/E and 3C/E: Instead of normalized data, it is important to show the real reactivation data. Are 1% or 50% of luciferase molecules reactivatable with Skd3 in this set up?

6) No direct evidence has been presented that the pore-loop Y430 makes direct contact with the substrate, hence it seems an overreach to claim that the conserved tyrosine binds the substrate. This statement should be qualified or better supported.

7) Some additional mutants should be tested before concluding that disaggregase activity correlates with disease severity (a term that needs to be defined). Otherwise, this claim should be toned down.

Updated Essential revisions in light of COVID-19:

1) Please address as many points as possible raised in the original summary.

2) Regarding Essential revision #1, above – the in vitro data can be strengthened by improved data analysis and discussion of the limitations, which should be possible without additional experiments. Specifically, the amount of luciferase as a % of total activity prior to aggregation should be reported to help support the idea that Skd3 is a robust disaggregase. Please also comment on the estimated fraction of α-synuclein present in fibrils in at the start of the in vitro disaggregation assay.

3) Regarding Essential revision #2, above – the fact that the in vivo experiments in Figure 7 do not distinguish between disaggregation and maintaining protein solubility should be directly addressed in the Discussion.

---

## [Author Response]

[…] Overall, the findings are of significant interest in addressing long-standing questions concerning the function of Skd3 and how metazoans handle mitochondrial aggregation in the absence of Hsp110 and Hsp78. This said, the study falls short on demonstrating true disaggregation activity in vitro and in vivo, behaviors that ideally would need to be validated before publication is warranted.

We respectfully disagree with the reviewers concerning the in vitro data. We have established that Skd3 is a potent disaggregase in vitro, capable of dissolving disordered aggregates and amyloid (please see further details below). We also establish that Skd3 maintains the solubility of many mitochondrial proteins in vivo, which is highly consistent with its potent disaggregase activity defined in vitro.

While this review was being finalized, the complexities of COVID-19 emerged. Given these recent developments, the reviewers recognize that the additional experiments they initially requested may not be possible. We would therefore ask that the authors first read the 'original' review for a sense of the issues raised by the reviewers, and then the 'updated' review, which follows for a revised set of requests.

We thank the reviewers for their thoughtful suggestions, which have helped us to improve the manuscript. We also appreciate the recognition of the difficulties in pursuing further experiments at this time during the pandemic. Indeed, at the time of finalizing this response (May 6^th^), it is not clear when UPenn will reopen labs (indeed Philadelphia is under stay at home orders until June 4^th^). Please find our response to both the original and updated reviews below.

Essential revisions:1) The authors demonstrate that Skd3 reactivates thermally inactivated luciferase but do not show directly that Skd3 actually dissociates luciferase aggregates.

We did not use thermally inactivated luciferase; rather, we used a long-established protocol (first developed in Susan Lindquist’s lab) for generating large chemically denatured, luciferase aggregates, which is a gold standard for the study of protein disaggregases (DeSantis et al., 2012; Glover and Lindquist, 1998). Thus, firefly luciferase was chemically denatured with 8M urea and then rapidly diluted 100-fold into a physiological buffer (DeSantis et al., 2012; Glover and Lindquist, 1998). This protocol yields aggregated luciferase species of predominantly ∼500-2,000kDa or greater in size and very few luciferase species smaller than ~400kDa can be detected (Glover and Lindquist, 1998). Importantly, these samples are devoid of misfolded, monomeric luciferase (Glover and Lindquist, 1998). Reactivation of luciferase in this assay is thus primarily achieved by the extraction and subsequent refolding of monomeric luciferase (M_w_~61kDa) from large aggregated structures (∼500-2,000kDa or larger) (Glover and Lindquist, 1998). Hence, in this assay, luciferase reactivation is an accurate proxy for luciferase disaggregation. We now emphasize this point in the subsection “Skd3 couples ATP hydrolysis to protein disaggregation and reactivation”.

Importantly, it has also been established that Hsp70 and Hsp40 are unable to disaggregate and reactivate luciferase found in these large aggregated structures (∼500-2,000kDa or larger), but can disaggregate and reactivate luciferase trapped in aggregated species smaller than 400kDa, albeit with a low yield (Glover and Lindquist, 1998). Indeed, only the combination of Hsp104, Hsp70, and Hsp40 can disaggregate and reactivate luciferase trapped in larger aggregates greater than ∼500kDa in size (Glover and Lindquist, 1998). Throughout our studies, we include the essential negative control of Hsp70 and Hsp40 in our luciferase disaggregation/reactivation assays. We would expect to see no reactivation of luciferase by Hsp70 and Hsp40 if luciferase is trapped in large aggregates. Indeed, because we see no reactivation of luciferase by the Hsp70 and Hsp40 control, we can be certain that (1) luciferase is predominantly trapped in large aggregated structures and (2) Skd3 or Hsp104, Hsp70, and Hsp40 disaggregate and reactivate luciferase trapped in these large aggregated structures. Thus, Skd3 or Hsp104, Hsp70, and Hsp40 activity in this assay cannot be explained by simple chaperone (i.e. inhibition of aggregation) activity or simple refolding of misfolded monomeric luciferase. Rather, reactivated luciferase was recovered by disaggregation from large aggregates. We now emphasize this point in the subsection “Skd3 couples ATP hydrolysis to protein disaggregation and reactivation”.

(The solubilization of α-synuclein in Figure 4 is insufficient to demonstrate such an activity because the experiments never start with the isolated fibrils only and no equivalent experiments are shown with luciferase).

The solubilization of a-synuclein fibrils shown in Figure 4 does in fact begin with isolated a-synuclein fibrils, which are devoid of soluble a-synuclein. That is, these samples are comprised of purely α-synuclein fibrils (assembled for 5 days at 37°C with agitation) and do not contain any soluble a-synuclein. We now clarify this point in the subsection “PARL-activated Skd3 dissolves α-synuclein fibrils”. The α-synuclein fibrils were acquired from Kelvin Luk’s lab (Luk et al., 2012). These α-synuclein fibrils have been characterized via electron microscopy and sedimentation assays to be large, fibrillar assemblies that drive thioflavin-S positive, Lewy body-like pathology and contingent neurodegeneration in mice (whereas monomeric α-synuclein does not) (Luk et al., 2012). Thus, they are a disease-causing entity. Furthermore, these a-synuclein fibrils react with thioflavin-T (an amyloid-diagnostic dye) in vitro and no soluble a-synuclein is present in the supernatant in the absence of ATP (no ATP buffer control) (Figure 4). Given that significantly more soluble a-synuclein was observed after Skd3 treatment compared to controls, this observation supports the conclusion that Skd3 disaggregates pre-formed α-synuclein fibrils and thus is a potent protein disaggregase.

An in vitro test for disaggregation activity (e.g., spinning down and quantifying aggregates as a function of Skd3 activity) is necessary to validate the primary point of the manuscript.

As noted above, we have provided these data for a-synuclein (Figure 4). Thus, we quantify the amount of soluble a-synuclein in the supernatant fraction after preformed a-synuclein fibrils are treated with Skd3. Our findings establish that Skd3 is able to liberate soluble forms of a-synuclein from preformed a-synuclein fibrils. In future studies, it will be of interest to pursue sedimentation-based assays of luciferase disaggregation (although as noted above, the recovery of reactivated luciferase is an accurate proxy for disaggregation under our specific assay conditions) or disaggregation of native Skd3 substrates (e.g. HAX1) in vitro to further validate Skd3 as a potent protein disaggregase.

2) Similar to (1), the authors should show that Skd3 is able to disaggregate proteins in vivo (the increased solubility of HAX1 in Figure 7 is insufficient to demonstrate such an activity because it cannot distinguish between whether Skd3 resolves aggregation or prevents aggregation from happening in the first place). One approach that might be of use is that of Janowsky et al., who showed that mitochondrial Hsp78 was to induce thermal aggregation of an imported protein in purified mitochondria and then monitor disaggregation +/- Hsp78 (von Janowsky et al., 2006). Other work by Liberek provides some alternative approaches.

The reviewers raise a good point. Our cell-based studies demonstrate the insolubility of specific substrates in the intermembrane space in the absence of Skd3. These data are highly consistent with Skd3 functioning as a protein disaggregase in vivo. However, they do not exclude the possibility that Skd3 might also function as a chaperone that prevents protein aggregation. In the text, we have been careful to discuss the effect of Skd3 on proteins in the cell as “changes in solubility” in the presence or absence of Skd3. We hope that our careful discussion sufficiently emphasizes the limitations of our experiments. Regardless, our findings establish a critical role for Skd3 in maintaining proteostasis in the mitochondrial intermembrane space.

In future studies, it will be of interest to assess Skd3 disaggregase activity in vivo using the approaches suggested by the reviewers. Indeed, experiments inspired by the work of Janowsky et al. would help to establish Skd3 disaggregase activity in vivo (Janowsky et al., 2006). Nevertheless, our findings strongly suggest that Skd3 is essential to maintain proteostasis in the mitochondrial intermembrane space and are highly consistent with Skd3 functioning as a protein disaggregase (which it can in vitro).

3) The changes in mitochondrial protein solubility observed in SKD3 knockout cells could be caused by loss of SKD3 disaggregase activity on specific proteins or secondary defects in protein import or membrane integration leading to accumulation of misfolded/uninserted proteins in the IMS. The authors should address this question in the Discussion.

The reviewers raise a good point. A section has been added to the Discussion to address these alternative explanations. It is possible that the changes in protein solubility are partially due to indirect effects of Skd3 on protein import into mitochondria or via sorting of membrane-targeted cargoes. We speculate that Skd3 might also play a role in protein import into mitochondria and further propose that Skd3 may play a role in maintaining the solubility of aggregation-prone mitochondrial targeting signals and membrane proteins during import and sorting (Endo et al., 1995; Poveda-Huertes et al., 2020; Yano et al., 2003). Nonetheless, our in vitro data strongly suggest that Skd3 can function as a potent protein disaggregase, and we suggest that it is likely to function in a similar manner in vivo. Indeed, Skd3 disaggregase activity is likely essential when proteins aggregate in the intermembrane space.

4) The loss of ATPase and disaggregase activity seen for the isolated NBD domain (SKD3NBD) could be explained by disassembly of SKD3 hexamers in the absence of the ankyrin domains. The authors should include information on whether these constructs form stable hexamers.

The reviewers raise an interesting point. We think it is likely that a lack of oligomerization of the isolated NBD domain contributes to the lack of ATPase activity. Previous studies highlighted in the text show that isolated NBD2 constructs of bacterial ClpB form dimers and monomers in vitro and are deficient for ATPase activity (Beinker et al., 2005). Indeed, size-exclusion traces from the purification of isolated Skd3 NBD shows what appears to be a primarily dimeric species. Further studies using SEC-MALS and DLS will be very useful in quantifying this observation. Added emphasis on this point has been made in the Results section of the text (subsection “Skd3 disaggregase activity requires the ankyrin-repeat domain and NBD”).

5) Figures 2C/E and 3C/E: Instead of normalized data, it is important to show the real reactivation data. Are 1% or 50% of luciferase molecules reactivatable with Skd3 in this set up?

The reviewers make a valuable point, which helps to strengthens the data presented in the manuscript. We have reanalyzed the data using a native luciferase standard curve for Figures 2C, E, 3C, E, and Figure 3—figure supplement 2D. In Figure 2C, Buffer alone had 0.04 ± 0.00% (± SEM) native luciferase activity, Hsp70 + Hsp40 reactivated 0.05 ± 0.00%, Hsp104 reactivated 0.08 ± 0.01%, Hsp104 + Hsp70 + Hsp40 reactivated 7.68 ± 0.44%, _MPP_Skd3 reactivated 2.87 ± 0.17%, and _MPP_Skd3 (no ATP) reactivated 0.04 ± 0.00%.

In Figure 2E, Buffer alone had 0.01 ± 0.00% native luciferase activity, Hsp70 + Hsp40 reactivated 0.02 ± 0.00%, Hsp104 reactivated 0.05 ± 0.00%, Hsp104 + Hsp70 + Hsp40 reactivated 6.71 ± 0.63%, _MPP_Skd3 reactivated 2.45± 0.32%, _MPP_Skd3^K387A^ reactivated 0.01 ± 0.00%, _MPP_Skd3^E455Q^ reactivated 0.01± 0.00%, and _MPP_Skd3^Y430A^ reactivated 0.01 ± 0.00%.

In Figure 3C, Buffer alone had 0.02 ± 0.00% native luciferase activity, Hsp70 + Hsp40 reactivated 0.03 ± 0.00%, Hsp104 reactivated 0.04 ± 0.01%, Hsp104 + Hsp70 + Hsp40 reactivated 8.58 ± 1.20%, _MPP_Skd3 reactivated 3.10± 0.44%, _MPP_Skd3 (no ATP) reactivated 0.02 ± 0.00%, _PARL_Skd3 reactivated 44.06± 3.78%, and _PARL_Skd3 (no ATP) reactivated 0.08 ± 0.01%.

In Figure 3E, Buffer alone had 0.02 ± 0.00% native luciferase activity, Hsp70 + Hsp40 reactivated 0.03 ± 0.00%, Hsp104 reactivated 0.08 ± 0.00%, Hsp104 + Hsp70 + Hsp40 reactivated 16.10 ± 1.66%, _PARL_Skd3 reactivated 38.92± 3.70%, _PARL_Skd3^K387A^ reactivated 0.02 ± 0.00%, _PARL_Skd3^E455Q^ reactivated 0.05± 0.02%, and _PARL_Skd3^Y430A^ reactivated 0.05 ± 0.00%.

To determine how much luciferase could be recovered using higher concentrations of _PARL_Skd3, we re-analyzed the data from Figure 3—figure supplement 2D. In Figure 3—figure supplement 2D, we found that 1.5µM _PARL_Skd3 was able to reactivate 75.60 ± 8.56% native luciferase and 3.0µM _PARL_Skd3 was able to reactivate 75.83 ± 6.80%. This information has been added to the text to supplement the figures and highlights the capability of Skd3 to disaggregate and reactivate luciferase. These findings also strongly suggest that Skd3 is capable of very effectively rescuing previously aggregated proteins.

6) No direct evidence has been presented that the pore-loop Y430 makes direct contact with the substrate, hence it seems an overreach to claim that the conserved tyrosine binds the substrate. This statement should be qualified or better supported.

The subsection “Skd3 couples ATP hydrolysis to protein disaggregation and reactivation” has been amended to qualify the statement and does not directly imply substrate contact from the pore loops. Rather, it frames them as pore-loop contacts and states that these residues are pivotal for translocating substrate in other HCLR clade AAA+ proteins (DeSantis et al., 2012; Gates et al., 2017; Lopez et al., 2020; Martin et al., 2008; Rizo et al., 2019). This issue has been corrected in the Results, Discussion, and legend for Figures 2 and 3.

7) Some additional mutants should be tested before concluding that disaggregase activity correlates with disease severity (a term that needs to be defined). Otherwise, this claim should be toned down.

The subsection “MGCA7-linked Skd3 variants display diminished disaggregase activity” has been modified with a short summary of the scoring system from the literature that defines the clinical severity of the disease (Pronicka et al., 2017). The experiments to test more 3-methylglutaconic aciduria, type VII (MGCA7)-linked variants were in progress. Unfortunately, due to COVID-19 they are now on hold. It will also be valuable in future studies to determine the effect of other MGCA7-linked mutations on Skd3 activity, both in vitro and in cells. Nonetheless, we have assessed the disaggregase and ATPase activity of four MGCA7-linked Skd3 variants and find that disaggregase activity is a better predictor of disease severity than ATPase activity.

Updated Essential revisions in light of COVID-19:1) Please address as many points as possible raised in the original summary.

Please see above.

2) Regarding Essential revision #1, above – the in vitro data can be strengthened by improved data analysis and discussion of the limitations, which should be possible without additional experiments. Specifically, the amount of luciferase as a % of total activity prior to aggregation should be reported to help support the idea that Skd3 is a robust disaggregase. Please also comment on the estimated fraction of α-synuclein present in fibrils in at the start of the in vitro disaggregation assay.

We discuss the controls and validation of the luciferase assays in Essential revision #1, which demonstrate that this assay accurately reports on disaggregase activity. In Essential revision #5, we quantify luciferase reactivation as % native luciferase activity and have added this information to supplement the figures in the text of the manuscript. We briefly summarize the analysis here: _MPP_Skd3 can recover ~3% native luciferase activity from the aggregates, _PARL_Skd3 can recover ~44% native luciferase activity from the aggregates and remarkably at higher concentrations, can recover ~75% native activity, which further supports the evidence of Skd3 possessing robust protein-disaggregase activity.

As noted in Essential revision #1, the α-synuclein fibrils are devoid of soluble a-synuclein. Indeed, there is negligible monomeric α-synuclein present and the sample is virtually 100% fibrillar α-synuclein. For example, there is virtually no signal in the supernatant in the no ATP buffer control (Figure 4).

3) Regarding Essential revision #2, above – the fact that the in vivo experiments in Figure 7 do not distinguish between disaggregation and maintaining protein solubility should be directly addressed in the Discussion.

This topic has been addressed above under Essential revision #2 and corrected in the text.